# Lie Symmetry Net: Preserving Conservation Laws in Modelling Financial Market Dynamics via Differential Equations

**Xuelian Jiang**                                                              *jiangxl133@nenu.edu.cn*
*School of Mathematics and Statistics*
*Northeast Normal University*

**Tongtian Zhu**                                                                  *raiden@zju.edu.cn*
*College of Computer Science and Technology*
*Zhejiang University*

**Yingxiang Xu**[*]                                                               *yxxu@nenu.edu.cn*
*School of Mathematics and Statistics*
*Northeast Normal University*

**Can Wang**                                                                        *wcan@zju.edu.cn*
*College of Computer Science*
*Zhejiang University*

**Yeyu Zhang**                                                       *zhangyeyu@mail.shufe.edu.cn*
*School of Mathematics*
*Shanghai University of Finance and Economics*

**Fengxiang He**                                                                    *F.He@ed.ac.uk*
*School of Informatics*
*University of Edinburgh*

**Reviewed on OpenReview:** *https://openreview.net/forum?id=rkfop9GyxB*

## Abstract

This paper employs a novel Lie symmetries-based framework to model the intrinsic symmetries within financial market. Specifically, we introduce *Lie symmetry net* (LSN), which characterises the Lie symmetries of the differential equations (DE) estimating financial market dynamics, such as the Black-Scholes equation. To simulate these differential equations in a symmetry-aware manner, LSN incorporates a Lie symmetry risk derived from the conservation laws associated with the Lie symmetry operators of the target differential equations. This risk measures how well the Lie symmetries are realised and guides the training of LSN under the structural risk minimisation framework. Extensive numerical experiments demonstrate that LSN effectively realises the Lie symmetries and achieves an error reduction of more than *one order of magnitude* compared to state-of-the-art methods. The code is available at https://github.com/Jxl163/LSN_code.

## 1 Introduction

A classic approach for modeling financial market dynamics is via stochastic differential equations (SDEs). Through the application of the Feynman-Kac formula (Del Moral & Del Moral, 2004), these SDEs can be transformed into corresponding Partial Differential Equations (PDEs), such as the Black-Scholes (BS) equation (Merton, 1973; Black & Scholes, 1973; Goodman & Stampfli, 2001; Rodrigo & Mamon, 2006). Traditionally,

---

[*]Corresponding author.

numerical methods such as Finite Volume Method (FVM) (Valkov, 2014) and B-spline collocation methods (Kadalbajoo et al., 2012; Huang & Cen, 2014) are used to simulate these equations. In recent years, AI - driven methodologies, exemplified by Physics - Informed Neural Networks (PINNs) (Raissi et al., 2019), have emerged as a powerful solution for solving differential equations by utilizing collocation data, a set of scattered points within the solution domain, to fit their dynamics.

A defining characteristic of SDEs is their "symmetry". A major family of mathematical tools to characterise the symmetry are Lie symmetry groups (Edelstein & Govinder, 2009; Paliathanasis et al., 2016; Gazizov & Ibragimov, 1998). Lie symmetries preserve the structural properties of solutions, simplifying the process of finding solutions and improving accuracy (Misawa, 2001; Kozlov, 2010; Marjanovic et al., 2015; Gaeta, 2017; Marjanovic & Solo, 2018). This principle is also applicable to the Black-Scholes (BS) equation in finance (Rao et al., 2016). Our vision is that the Lie symmetries can represent some intrinsic symmetry in financial markets, though in an abstract manner (Kozlov, 2010).

However, Lie symmetries remain largely unexplored in current AI-driven DE solvers. Neglecting this symmetry may lead AI-driven approaches to learn solutions performing well on training collocation data, yet fail to satisfy inherent structural constraints. This is caused by the imbalance or other limitations, such as the low quality of collocation data, which may compromise the generalizability of the learned solver. As a result, the performance of the learned solver on unseen data remains uncertain, as widely reported in the literature (Cohen et al., 2021; Yang et al., 2023; Li et al., 2024).

This paper aims to answer the following fundamental question:

> *Could Lie symmetries facilitate AI-driven DE solvers in simulating financial market dynamics, and how?*

Motivated by this question, we design Lie symmetry net (LSN), which enables the simulation of financial market dynamics while preserving Lie symmetries.

Similar to many symmetries in physics, the Lie symmetries can be transformed into conservation laws (Kara & Mahomed, 2002; Edelstein & Govinder, 2009; Khalique & Motsepa, 2018; Özkan et al., 2020). Specifically, for the Black-Scholes, the conservation laws derived from Lie symmetries are

$$D_t T^t + D_x T^x = 0,$$

where $D_t$, $D_x$ represents the partial derivative with respect to time $t$ or asset price $x$, and $(T^t, T^x)$ represents the conservation vector subject to the symmetry condition (*i.e.*, Lie point symmetry operator) $G$, such that the action of $G$ on the conservation vector satisfies $G(T^t, T^x) = 0$ (Kara & Mahomed, 2000; Edelstein & Govinder, 2009).

In our LSN, we design a novel *Lie conservation residual* to quantify how well the Lie symmetries are realised on one specific point in the collocation data space that comprises asset price and time. This Lie conservation residual then induces a *Lie symmetry risk* that aggregates the residual over the collocation data space, and thus characterises how Lie symmetries are realised from a global view. It is worth noting that this Lie symmetry risk depends on the specific conservation law, and thus the specific Lie symmetry operator. This Lie symmetry risk is then integrated with risk functions measuring how well the LSN fits the collocation data (Raissi et al., 2019; Xie et al., 2023), and formulates the structural risk of LSN. We can optimise the LSN under the structural risk minimisation (SRM) framework (Shawe-Taylor et al., 1998) to learn an differential equations solver while preserving the Lie symmetries.

Extensive numerical experiments are conducted to verify the superiority of LSN. We compare LSN with state-of-the-art methods including IPINNs (Bai et al., 2022), sfPINNs (Wong et al., 2022), ffPINNs (Wong et al., 2022) and LPS (Akhound-Sadegh et al., 2024). The results demonstrate that LSN consistently outperforms these methods, achieving error reductions of more than an order of magnitude. Specifically, the error magnitude with single operator reaches $10^{-3}$, while with combined operators (refer to Section 5.3), it further decreases to $10^{-4}$.

The paper is structured as follows. Section 2 provides an overview of related work. Section 3 discusses the background of PINNs and SDEs. Section 4 introduces the methodology of LSN. Section 5 presents numerical

experiments to validate the effectiveness of LSN. Finally, Section 6 draws conclusions and outlines directions for future research. Appendix A provides additional background, models and the theoretical analyses of LSN.

## 2 Related Works

**Numerical Equation Solvers**. Numerical methods have long been essential for solving partial differential equations in various domains, including financial market modeling. Significant progress has been made in this area with models such as the Black-Scholes equation. Traditional approaches, including finite volume methods (FVM) (Valkov, 2014) and B-spline collocation methods (Kadalbajoo et al., 2012; Huang & Cen, 2014), have been widely applied to solve these equations (Koc et al., 2003; Rao et al., 2016). These grid-based techniques rely on discretizing the spatial and temporal domains, transforming the continuous equations into discrete problems suitable for simulation. However, these methods often come with high computational complexity, which may limit their applicability.

**Neural Equation Solvers**. In recent years, there has been a gradual increase in applying neural networks to solve differential equations. Two main approaches have emerged in this area. The first one, neural operator methods (Li et al., 2020; Lu et al., 2021; Kovachki et al., 2023; Hao et al., 2023), focuses on learning the mapping between the input and output functions of the target equations. In contrast, the second approach, Physics-Informed Neural Networks (PINNs) (Raissi et al., 2019), directly approximates the solution of the equations, rather than relying exclusively on the values of the function derived from collocation data. PINNs and their variants, such as sfPINNs (Wong et al., 2022) and ffPINNs (Wong et al., 2022), have gained popularity for utilizing physical laws into the training process. Recent studies have successfully applied PINNs to solve financial equations, introducing efficient methods like IPINNs (Bai et al., 2022), which incorporate regularization terms for slope recovery.

**Input Space Symmetries**. Despite significant advancements in machine learning, the inherent symmetries within input data remain underutilized. Equivariant Neural Networks (ENNs) represent an innovative architectural approach explicitly designed to encode and utilize these symmetries, resulting in the property of equivariance (Celledoni et al., 2021; Satorras et al., 2021; Gerken et al., 2023). This property enables ENNs to efficiently model data with structured invariances. In the domain of computer vision, ENNs have shown remarkable efficacy over traditional neural networks. For example, in image classification tasks, they achieve higher accuracy in recognizing objects subjected to transformations such as rotation or translation (Rojas-Gomez et al., 2024). By preserving the symmetries of input data, ENNs retain critical geometric information during feature extraction (He et al., 2021). Moreover, ENNs also exhibit exceptional performance in physical simulations by effectively modeling symmetrical transformations of physical systems (Bogatskiy et al., 2024), thereby improving both the accuracy and computational efficiency of simulations.

**Parameter Space Symmetries**. Over recent decades, a range of studies have analyzed symmetries in neural network parameter spaces—transformations of network parameters that leave the underlying network function unchanged (Hecht-Nielsen, 1990; Sussmann, 1992). Notable examples of such symmetries include invertible linear transformations in linear networks and rescaling transformations in homogeneous networks (Badrinarayanan et al., 2015; Du et al., 2018). Recent studies further provide deeper insights and expand the scope of parameter space symmetries. Zhao et al. (2023) investigate diverse parameter space symmetries and derive novel, nonlinear, data-dependent symmetries. Ziyin (2024) establish a systematic framework for analyzing symmetries, showing that rescaling symmetry induces sparsity, rotation symmetry leads to low-rank structures, and permutation symmetry facilitates homogeneous ensembling. Further, Ziyin et al. (2024) examines how exponential symmetries, a broad subclass of continuous symmetries, interplay with stochastic gradient descent.

**Lie symmetries**. While input space and parameter space symmetries have played crucial roles in neural network design, they have rarely been applied to characterize the intrinsic symmetries of differential equations. Lie symmetry analysis, in contrast, provides a systematic framework for analyzing these symmetries. It represents a significant type of symmetry with widespread applications in mathematics (Olver, 1993; Ibragimov, 1995). As a powerful tool for solving partial differential equations (PDEs), Lie symmetry method leverages the symmetry group of an equation to reduce its order or transform it into a simpler, more tractable form,

thereby simplifying the solution process (Gazizov & Ibragimov, 1998; Liu et al., 2009; Oliveri, 2010). Despite its well-established theoretical foundation in traditional mathematical fields, its integration into neural DE solvers remains largely unexplored.

A more recent work by Akhound-Sadegh et al. (2024) proposes to incorporate Lie symmetries into PINNs by minimizing the residual of the determining equations of Lie symmetries. While this approach offers an interesting direction, our LSN follows a different methodology. Specifically, their Lie point symmetry (LPS) method focuses on minimizing these symmetry residuals, whereas our LSN realises Lie symmetries by preserving the conservation laws derived from the Lie symmetry operators. These conservation laws are fundamental principles inherent to the system described by the differential equations. Additionally, LPS has been validated only on the Poisson and Burgers equations in their original paper and its effectiveness in leveraging inherent symmetries in financial markets remains unclear. In contrast, our comprehensive comparative experiments in financial domain, specifically on the BS equation across various parameters, clearly demonstrate the superiority of LSN over LPS by reducing testing error by an order of magnitude.

## 3  Preliminaries

This section provides the essential background knowledge. We begin with an introduction to Physics-Informed Neural Networks. We then cover stochastic differential equations and explain how the Feynman-Kac formula allows for the transformation of a SDE into a corresponding PDE. To concretize this theoretical framework, we provide illustrative examples including the Black-Scholes equation. For further terminology and models related to finance and Lie symmetries, such as the Vašiček equation, please refer to Appendix A.

### 3.1  Physics-Informed Neural Networks (PINNs)

PINNs integrate physical information from the equations to approximate numerical solutions of PDEs, rather than relying exclusively on collocation data. The use of PINNs to solve differential equations typically begins with generating a collocation dataset $\mathcal{S} = \left\{ \{(x_i^n, t_i^n)\}_n^{N_i}, \{(x_s^n, t_s^n)\}_n^{N_s}, \{x_t^n\}_n^{N_t} \right\}$ by randomly sampling points within the solution domain. To understand how PINNs operate, consider the following evolution partial differential equation,

$$\begin{cases} \frac{\partial u(x,t)}{\partial t} = \mathbf{L}[u] & \forall (x,t) \in \Omega \times [0, T], \\ u(x,0) = \varphi(x) & \forall x \in \Omega, \\ u(y,t) = \psi(y,t) & \forall (y,t) \in \partial\Omega \times [0, T], \end{cases} \tag{1}$$

where $\mathbf{L}[u]$ represents a differential operator, $\Omega$ denotes a bounded domain, $\varphi(x)$ and $\psi(y,t)$ correspond to the initial and boundary conditions, respectively, $T$ refers to the terminal time, and $u(x,t)$ is the function to be determined. To solve Equation (1), PINNs approximate the exact solution by modeling $u$ as a neural network $\hat{u}$ and minimizing an empirical loss function.

$$\hat{\mathcal{L}}_{PINNs} = \frac{1}{N_i} \sum_{n=1}^{N_i} \left| \frac{\partial \hat{u}_\theta(x_i^n, t_i^n)}{\partial t} - \mathbf{L}[\hat{u}](x_i^n, t_i^n) \right|^2 + \frac{1}{N_b} \sum_{n=1}^{N_b} |\hat{u}_\theta(x_b^n, t_b^n) - \psi(x_b^n, t_b^n)|^2 + \frac{1}{N_t} \sum_{n=1}^{N_t} |\hat{u}_\theta(0, x_t^n) - \varphi(x_t^n)|^2 ,$$

where $\theta$ denotes the network parameters. The first term evaluates the residual of the PDE, while the subsequent terms quantify the errors associated with the boundary and initial conditions, respectively.

### 3.2  Stochastic Differential Equation (SDE)

SDE (Øksendal & Øksendal, 2003) provide a mathematical framework for modeling systems influenced by random disturbances. To understand the dynamics of a stochastic process $X_t$, we consider the SDE of the following general form

$$dX_t = \mu(X_t, t)dt + \sigma(X_t, t)dW_t, \tag{2}$$

where $X_t$ represents the stochastic variable of interest and $W_t$ is a standard Wiener process (as defined in Definition A.1). The functions $\mu(X_t, t)$ and $\sigma(X_t, t)$, known as the drift and diffusion coefficients, respectively, are functions that characterise the deterministic and stochastic components of the dynamics.

The Feynman-Kac formula provides a critical theoretical framework to establish a connection between certain types of PDEs and SDEs (see Definition A.6). We illustrate this with a representative example from finance.

**Example** (Black-Scholes equation (Janson & Tysk, 2006)). Considering a frictionless and arbitrage-free financial market comprising a risk-free asset and a unit risky asset, the dynamics of the market can be modeled by the following SDE:

$$dx_t = rx_t dt + \sigma x_t dW_t, \tag{3}$$

where $x$ denotes the price of a unit risky asset, $t$ represent time, $\sigma$ is the volatility, $r$ is the risk-free interest rate and $W_t$ is a standard Wiener process (refer to Definition A.1). Applying the Feynman-Kac formula, the Black-Scholes equation for evaluating the price $u(x, t)$ of a European call option (refer to Definition A.2) is derived as follows:

$$\begin{cases} \frac{\partial u}{\partial t} + \frac{1}{2}\sigma^2 x^2 \frac{\partial^2 u}{\partial x^2} + rx\frac{\partial u}{\partial x} - ru = 0 & \Omega \times [0, T], \\ u(x, T) = \max(x - K, 0) & \Omega \times T, \\ u(0, t) = 0 & \partial\Omega \times [0, T], \end{cases} \tag{4}$$

where $K$ is the strike price, $T$ is the expiry time of the contract and $\Omega$ is a bounded domain. The solution to this equation can be written as follows (Bai et al., 2022):

$$u(x, t) = x\mathcal{N}(d_1) - K\exp^{-r(T-t)}\mathcal{N}(d_2), \, d_1 = \frac{ln(x/K) + (r + 0.5\sigma^2)(T-t)}{\sigma T - t}, \, d_2 = d_1 - \sigma\sqrt{T-t},$$

where $\mathcal{N}$ denotes the standard normal distribution.

### 3.3 Lie Group Analysis

Groups, which mathematically characterize symmetries, describe transformations that preserve certain invariances. Formally, a group $(G, \cdot)$ is defined as a set $G$ equipped with a binary operation $\cdot$ that satisfies the properties of associativity, contains an identity element $e \in G$, and ensures the existence of an inverse element $g^{-1}$ for each $g \in G$ (Rotman, 2012). When groups are equipped with the structure of differentiable manifolds, they are referred to as Lie groups and play a fundamental role in the analysis of continuous symmetries.(Knapp & Knapp, 1996). Lie group analysis provides a powerful tool for studying symmetry, conservation laws, and dynamic systems of equations (Gazizov & Ibragimov, 1998; Paliathanasis et al., 2016; Brandstetter et al., 2022; Otto et al., 2023; Dalton et al., 2024; Yang et al., 2024). The goal of Lie group analysis is to identify the symmetries of an equation, especially those transformations under Lie group actions that leave the equation invariant. These symmetries facilitates conservation law derivation (Edelstein & Govinder, 2009), simplifies solutions, and lowers computational complexity (Rao et al., 2016; Khalique & Motsepa, 2018).

## 4 Lie Symmetry Net

In this section, we introduce Lie symmetry net (LSN). In Section 4.1, we briefly derive the corresponding conservation law from the Lie symmetry operators of the target equations, which in turn lead to the Lie symmetry risk of LSN. In Section 4.2, we discuss the structure risk minimization of Lie Symmetry Net based on the Lie symmetry risk.

### 4.1 Lie Symmetries in Equations

This subsection presents the Lie symmetry operators for the Black-Scholes equation, and derive the corresponding conservation laws, which allows us to define the associated Lie symmetry risk.

**Lie Symmetry Operator.** Lie symmetry operator is a major mathematical tool for characterizing the symmetry in PDEs (see Definition A.3) (Paliathanasis et al., 2016). The Lie symmetry operators (Gazizov &

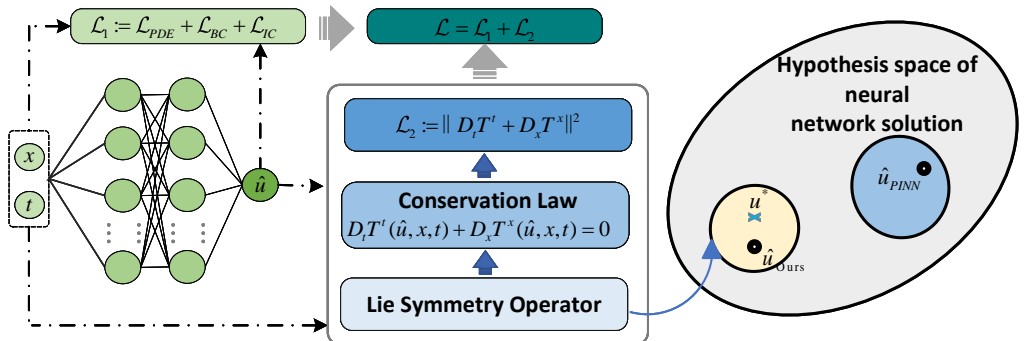

Figure 1: Schematic Diagram of Lie Symmetry Network Architecture. **Left:** The incorporation of a Lie symmetries blocks into a PINN architecture. **Right:** The hypothesis space of PINN, denoted as $\hat{u}_{PINN}$ (blue circle), is a subset of the broader neural network solution space ( gray circle). $u^*$ represents the exact solution. The yellow circle signifies the hypothesis space that satisfies symmetry conditions. The incorporation of Lie symmetries blocks produces our solution $\hat{u}_{ours}$. In simpler terms, being closer to the space where the exact solution exists leads to fewer errors. The non-intersection between "$\hat{u}_{PINN}$" and "$\hat{u}_{Ours}$" in the figure is a specific case provided to enhance clarity in our exposition.

Ibragimov, 1998; Edelstein & Govinder, 2009) of BS Equation (4) are given by the vector field

$$
\begin{aligned}
G_\phi &= \phi(t,x)\frac{\partial}{\partial u}, \, G_1 = \frac{\partial}{\partial t}, \, G_2 = x\frac{\partial}{\partial x}, G_3 = u\frac{\partial}{\partial u}, \\
G_4 &= 2t\frac{\partial}{\partial t} + (\ln x + Zt)x\frac{\partial}{\partial x} + 2rtu\frac{\partial}{\partial u}, G_5 = \sigma^2 tx\frac{\partial}{\partial x} + (\ln x - Zt)u\frac{\partial}{\partial u}, \\
G_6 &= 2\sigma^2 t^2\frac{\partial}{\partial t} + 2\sigma^2 tx\ln x\frac{\partial}{\partial x} + \left((\ln x - Zt)^2 + 2\sigma^2 rt^2 - \sigma^2 t\right)u\frac{\partial}{\partial u},
\end{aligned}
\tag{5}
$$

where $Z = r - \sigma^2/2$, and $\phi(t,x)$ is an arbitrary solution to Equation (4) without any boundary condition or initial condition. The first symmetry $G_\phi$ is an infinite-dimensional symmetry, arising as a consequence of linearity. These Lie symmetry operators form an infinite Lie group vector space (Edelstein & Govinder, 2009). The derivation of Lie symmetry operators for general differential equations is included in Appendix A.3.

These Lie symmetry operators not only provide a deeper insight into the structure of the PDEs but also form the foundation for deriving conservation laws associated with these equations.

**Conservation Law.** Similar to many symmetries in physics, the Lie symmetries can be transformed to conservation laws (Kara & Mahomed, 2002; Edelstein & Govinder, 2009; Khalique & Motsepa, 2018; Özkan et al., 2020). In this paper, we interpret the Lie symmetry point operators as the following conservation laws: regardless of how the space $x$, time $t$, and exact solution $u$ vary, the conservation vector $(T^t, T^x)$ corresponding to the Lie point symmetry remains zero, *i.e.*,

$$
D_t T^t(u, x, t) + D_x T^x(u, x, t) = 0,
\tag{6}
$$

where the $D_t$, $D_x$ represents the partial derivative with respect to time $t$ or space $x$, and $(T^t, T^x)$ represents the conservation vector subject to the symmetry condition (*i.e.*, Lie point symmetry operator) $G$, such that the action of $G$ on the conservation vector satisfies $G(T^t, T^x) = 0$ (Edelstein & Govinder, 2009). It is worth noting that this Lie symmetry risk depends on the specific conservation law. The derivation of the conservation law for general differential equations is included in Appendix A.4.

We can derive the conservation law of the operator $G_2$ (see Equation (5)) of BS equation as follows (Edelstein & Govinder, 2009):

$$
\begin{cases}
T_2^t(u,x,t) = -\dfrac{\partial u}{\partial x}l(t) + \dfrac{\mathscr{A}}{x} + \dfrac{2\mathscr{B}u}{\sigma^2 x}\mathrm{e}^{-rt}, \\[2ex]
T_2^x(u,x,t) = \dfrac{\partial u}{\partial t}l(t) + u\dfrac{\partial l(t)}{\partial t} + g(t) - \mathscr{B}u\mathrm{e}^{-rt} + \mathscr{B}\left(\dfrac{\partial u}{\partial x} + \dfrac{2ru}{\sigma^2 x}\right)x\mathrm{e}^{-rt},
\end{cases}
\tag{7}
$$

where $\mathscr{A}$ and $\mathscr{B}$ are arbitrary constants, and $l(t)$ and $g(t)$ are arbitrary functions with respect to $t$. Unless stated otherwise, consider $\mathscr{A} = \mathscr{B} = 1$, $l(t) = t$, and $g(t) = t^2$. The selection of $G_2$ is primarily motivated by its simplicity, as it effectively illustrates the fundamental concepts of the proposed method. Additional experiments in Section 5.3 demonstrate the effectiveness of the proposed methods when applied with various operators and their combinations on the Vašiček equation.

We can then define the Lie conservation residual $\mathcal{R}_{Lie}$ according to Equation (6) to evaluate the extent to which the Lie symmetries are realised at a specific point in the collocation data space.

**Definition 4.1** (Lie conservation residual). Combining the Lie symmetry operator and the conservation law, we define the Lie conservation residual of $\hat{u}$ as follows:

$$
\mathcal{R}_{Lie}[\hat{u}] = D_t T^t(\hat{u}) + D_x T^x(\hat{u}),
\tag{8}
$$

where the $D_t$, $D_x$ represents the partial derivative with respect to $t$ or $x$, and $(T^t, T^x)$ represents the conservation vector subject to the symmetry condition $G$.

We can aggregate the Lie symmetries residuals over the entire collocation data space to obtain the Lie symmetry risk, which characterises the degree to which the Lie symmetries are realised from a global perspective.

**Definition 4.2** (Lie symmetry risk). According to the expression in Equation (8), the definition of Lie symmetry risk is provided as follows:

$$
\mathcal{L}_{Lie}[\hat{u}_\theta](x,t) = \int_{\Omega \times [0,T]} |\mathcal{R}_{Lie}[\hat{u}_\theta](x,t)|^2 \, dxdt,
$$

where $\hat{u}_\theta$ denotes the network output, and $\theta$ denotes the network parameters.

*Remark* 4.3. The Lie symmetry risk $\mathcal{L}_{Lie}$ focuses solely on learning the symmetry of the problem without taking into account the underlying physical laws of the problem.

This Lie symmetry risk is defined over the collocation data distribution, which is unknown in practice. We thus define the *Empirical Lie symmetry risk* $\hat{\mathcal{L}}_{Lie}$ as an approximation of the Lie symmetry risk $\mathcal{L}_{Lie}$,

**Definition 4.4** (Empirical Lie symmetry risk). Summing up the Lie symmetry operators at $N_i$ discrete points provides an approximation to the Lie symmetry risk.

$$
\hat{\mathcal{L}}_{Lie}(\theta, \mathcal{S}) := \frac{1}{N_i} \sum_{n=1}^{N_i} |\mathcal{R}_{Lie}[\hat{u}_\theta](x_i^n, t_i^n)|^2,
\tag{9}
$$

where $\mathcal{S} = \{(x_i^n, t_i^n)\}_{n=1}^{N_i}$ represents the set of these $N_i$ discrete points.

## 4.2 Structure Risk Minimisation

In this subsection, we present the structure risk minimization of LSN based on the Lie symmetry risk. For clarity, we reformulate the Black-Scholes Equation (4) to match the format of Equation (1) (see Appendix A.2), aligning it with the Vašíček Equation (23) in the Appendix A.3 for a coherent presentation, where

$$
\mathbf{L}[u] := \frac{1}{2}\sigma(x)^2 \frac{\partial^2 u(x,t)}{\partial x^2} + \mu(x)\frac{\partial u(x,t)}{\partial x} + \upsilon(x)u(x,t),
\tag{10}
$$

is a differential operator with respect to three bounded affine functions $\sigma(x)$, $\mu(x)$ and $\upsilon(x)$ (for BS equation: $\sigma(x) = \sigma x$, $\mu(x) = rx$, $\upsilon(x) = -r$ and $\varphi(x) = max(x - K, 0)$ (for Vašiček Equation (23) $\sigma(x) = \sqrt{2\alpha} = \sigma$, $\mu(x) = \lambda(\beta - x) = -x$, $\upsilon(x) = \gamma x$ and $\varphi(x) = 1$).

**Data Fitting Residuals.** The following functions $\mathcal{R}_j$ $(j = \{i, s, t\})$ characterise how well the LSN is fitting the collocation data according to Equation (1), for $\forall \hat{u} \in C^2(\mathbb{R}^d)$

$$
\begin{array}{ll}
\mathcal{R}_i[\hat{u}](x,t) = \frac{\partial \hat{u}(x,t)}{\partial t} - \mathbf{L}[\hat{u}](x,t) & (x,t) \in \Omega \times [0,T], \\
\mathcal{R}_s[\hat{u}](y,t) = \hat{u}(y,t) - \psi(y,t) & (y,t) \in \partial\Omega \times [0,T], \\
\mathcal{R}_t[\hat{u}](x) = \hat{u}(0,x) - \varphi(x) & x \in \Omega.
\end{array}
\tag{11}
$$

We define the empirical loss function $\hat{\mathcal{L}}1$ to approximate the population risk $\mathcal{L}1$ in fitting the collocation data, by utilizing the aforementioned residuals, as follows:

$$
\begin{aligned}
\hat{\mathcal{L}}_1[\hat{u}_\theta](x,t) &= \hat{\mathcal{L}}_{PDE}[\hat{u}_\theta](x,t) + \hat{\mathcal{L}}_{BC}[\hat{u}_\theta](x,t) + \hat{\mathcal{L}}_{IC}[\hat{u}_\theta](x,t) \\
&:= \frac{1}{N_i} \sum_{n=1}^{N_i} |\mathcal{R}_i[\hat{u}_\theta](x_i^n, t_i^n)|^2 + \frac{1}{N_s} \sum_{n=1}^{N_s} |\mathcal{R}_s[\hat{u}_\theta](x_s^n, t_s^n)|^2 + \frac{1}{N_t} \sum_{n=1}^{N_t} |\mathcal{R}_t[\hat{u}_\theta](x_t^n)|^2.
\end{aligned}
$$

Here, $\mathcal{S} = \left\{ \{(x_i^n, t_i^n)\}_n^{N_i}, \{(x_s^n, t_s^n)\}_n^{N_s}, \{x_t^n\}_n^{N_t} \right\}$ denotes the collocation data set, which is generated by sampling points within the domain $\Omega \times [0,T]$ according to a Gaussian distribution.

*Remark* 4.5. $\hat{\mathcal{L}}_1$ essentially corresponds to the Physics-Informed Neural Networks (PINNs) (please refer to Section 3), which accurately estimates the majority of the training collocation data based on the inherent physical laws of the problem. However, it does not explicitly consider the symmetry within.

**Structural Risk of LSN.** We next present an empirical approximation of the structural risk $\mathcal{E}(\theta)$ of LSN.

**Definition 4.6** (Empirical structural risk of LSN)**.** The empirical loss of LSN is defined as follows

$$
\hat{\mathcal{E}}(\theta, \mathcal{S}) := \lambda_1 \hat{\mathcal{L}}_{PDE}(\theta, \mathcal{S}_i) + \lambda_2 \hat{\mathcal{L}}_{BC}(\theta, \mathcal{S}_s) + \lambda_3 \hat{\mathcal{L}}_{IC}(\theta, \mathcal{S}_t) + \lambda_4 \hat{\mathcal{L}}_{Lie}(\theta, \mathcal{S}_i),
\tag{12}
$$

where $\mathcal{S} = \left\{ \mathcal{S}_i \{(x_i^n, t_i^n)\}_n^{N_i}, \mathcal{S}_s \{(x_s^n, t_s^n)\}_n^{N_s}, \mathcal{S}_t \{x_t^n\}_n^{N_t} \right\}$ are the training collocation data sets, and $\lambda_i$ $(i = 1, \ldots, 4)$ denote the hyperparameters.

We train LSN by solving the following minimisation problem,

$$
\theta^* = \arg\min_\theta \lambda_1 \hat{\mathcal{L}}_{PDE}(\theta, \mathcal{S}_i) + \lambda_2 \hat{\mathcal{L}}_{BC}(\theta, \mathcal{S}_s) + \lambda_3 \hat{\mathcal{L}}_{IC}(\theta, \mathcal{S}_t) + \lambda_4 \hat{\mathcal{L}}_{Lie}(\theta, \mathcal{S}_i),
$$

and the minimum $\hat{u}_{\theta^*}$ corresponds to the well-trained LSN.

# 5 Experiments

In this section, we conduct three main experiments. In Section 5.1, we perform an ablation study on LSN, comparing it to PINNs (Raissi et al., 2019) under varying equation parameters. Next, we compare LSN with state-of-the-art baselines, including IPINNs (Bai et al., 2022), sfPINNs (Wong et al., 2022), ffPINNs (Wong et al., 2022), and LPS (Akhound-Sadegh et al., 2024), thereby further validating its superiority. In Section 5.2, we extend LSN to multiple models, providing experimental comparisons under both single-operator and combined-operator settings to verify its effectiveness. In Section 5.3, we test LSN on real financial data from Yahoo Finance to demonstrate its practical applicability.

We start by providing a brief introduction to the parameter settings for all experiments, with specific parameters for each experiment detailed in their respective sections.

**Data.** The small-scale experiments employ a training set 50 internally scattered points and 2000 points randomly placed at the boundaries, while the test set consists of 2,500 (or 200) uniformly sampled points. The

Table 1: Experimental Parameter Configuration. Values in parentheses represent weights for the enlarged collocation data set, while values not in parentheses represent weights for the small collocation data set.

| RFR (r) | volatility ($\sigma$) | weight | | | | learning rate ($lr$) | iteration |
|---|---|---|---|---|---|---|---|
| | | $\lambda_1$ | $\lambda_2$ | $\lambda_3$ | $\lambda_4$ | | |
| 0.1 | 0.4 | 0.001(0.001) | 1(1) | 0.1(0.1) | 1(0.1) | 0.001 | 200,000 |
| 0.1 | 0.5 | 0.001(0.0001) | 1(1) | 0.1(0.1) | 10(0.01) | 0.001 | 200,000 |
| 0.11 | 0.4 | 0.001(0.001) | 1(1) | 0.1(0.1) | 1(1) | 0.001 | 200,000 |

large-scale experiments employ a training set of 2000 internally scattered points and 8000 points randomly placed at the boundaries, while the test set consists of 2,500 (or 200) uniformly sampled points.

**Neural Architecture and Optimiser.** The LSN network employs a fully connected architecture, consisting of 9 layers with each layer having a width of 50 neurons. The tanh function is used as the activation function. For optimization, we choose Adam with an initial learning rate of 0.001 and a learning rate decay factor $\Gamma$.

**Equation Parameters**. For the parameters of the Black-Scholes equation, we set them to $K = 10$, $x \in [0, 20]$, and $t \in [0, 1]$, following the conventions established in the existing literature (Bai et al., 2022).

**Evaluation.** The evaluation metrics include relative test error (refer to the definition in Section Definition A.5) and conservation error $\hat{\mathcal{L}}_{Lie}$.

## 5.1 Experiments on Black-Scholes Equation

In this section, we perform experiments focused on the Black-Scholes (BS) equation to conduct ablation studies and comparative analyses on LSN. In Section 5.1.1, we perform ablation studies on LSN with both small- and large-scale datasets, comparing it to the baseline PINN method, to demonstrate its performance improvements across varying data scales. In Section 5.1.2, we compare LSN with several state-of-the-art methods, including the Fourier frequency-based ffPINN and sfPINN, the BS-equation-specific IPINN, and the Lie symmetry-based LPS, to comprehensively assess its performance.

### 5.1.1 Comparison with PINNs

**Experimental Design.** We conduct comparative experiments between LSN and baseline PINNs under the following four sets of hyperparameter setups.

For the first configuration, we chose $r = 0.1$, $\sigma = 0.05$, a learning rate decay rate $\Gamma = 0.99$, and $Iterations = 50,000$, the weight for LSN's loss function was set as $\lambda_1 = 0.001$, $\lambda_2 = 1 - \lambda_1 = 0.999$, $\lambda_3 = 0.001$ and $\lambda_4 = 0.001$, while for PINNs, the weight was set as $\lambda_1 = 0.001$, $\lambda_2 = 0.999$ and $\lambda_3 = 0.001$. For the second configuration, we select $r = 0.1$, $\sigma = 0.2$, a learning rate decay rate of $\Gamma = 0.95$, and $Iterations = 90,000$. The weights for the loss functions remains the same as the first configuration. For the third setting, the experimental parameter settings for the small collocation dataset under other parameters are given in the Table 1. Regarding the fourth setting, the experimental parameter settings for the enlarged collocation data set are provided in the Table 1, with values in parentheses.

We visualize the numerical solutions obtained on the test set using these two sets of hyperparameters in Figure 2. In identical experimental configurations, LSN outperforms PINNs, achieving a point-wise error magnitude of $10^{-2}$ in contrast to $10^{-1}$ observed with PINNs.

The error curves for LSN and PINNs with respect to the number of training steps for the third set of parameters are presented in Figure 3. It can be observed from Figure 3 that the conservation error of PINNs is of the order $10^{-1}$, whereas LSN can achieve an error on the magnitude $10^{-4}$. Furthermore, the relative error of LSN also consistently remains lower than that of PINNs.

In the fourth set of experiments, we increase the number of collocation data points to 10k. It can be observed from Figure 4 that increasing the number of collocation data points improves the accuracy of both PINNs

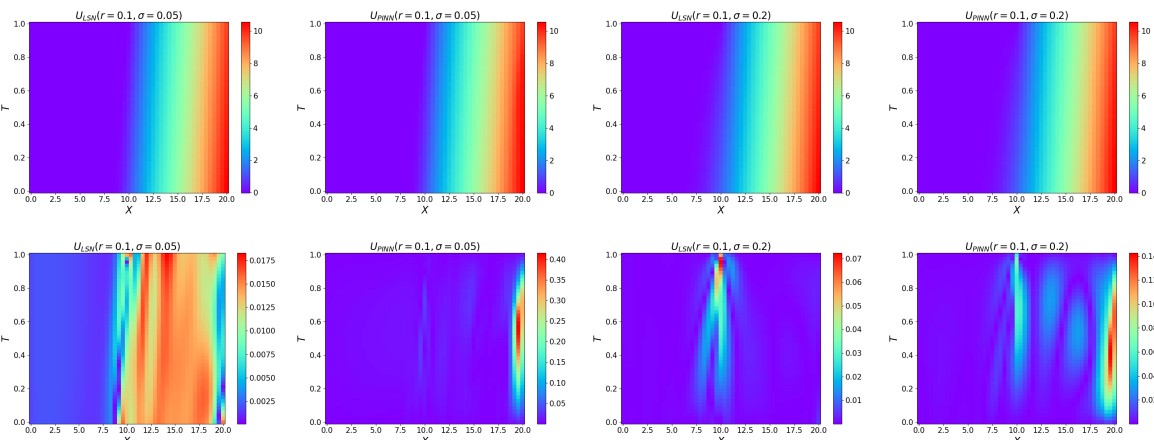

Figure 2: Visual representations of numerical solutions obtained using LSN and PINNs, along with absolute errors compared to the exact solution. The configuration of parameters is as follows: (1) $r = 0.1$, $\sigma = 0.05$, $\Gamma = 0.99$, and $Iterations = 50,000$ (the left two column); (2) $r = 0.1$, $\sigma = 0.2$, $\Gamma = 0.95$, and $Iterations = 90,000$ (the right column).

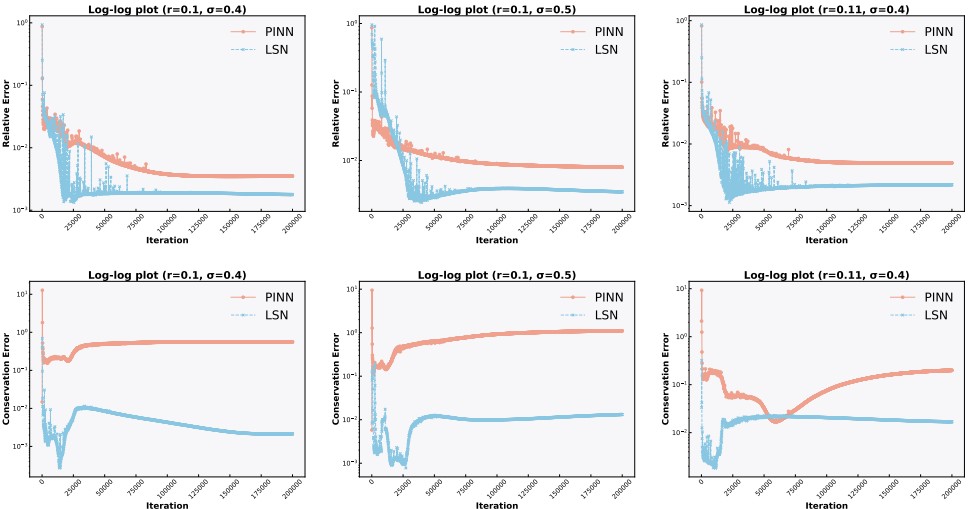

Figure 3: Log-log relative test error curves of PINNs and LSN under the third parameters configuration.

Table 2: Comparisons of relative test error of LSN and PINNs after 80,000 training iterations. Here $\Gamma$ represents the rate of learning rate decay, while "Factor" represents the ratio of the test error of PINNs to that of LSN.

| RFR (r) | Volatility ($\sigma$) | Relative test error | | Factor |
|---|---|---|---|---|
| | | PINNs | LSN | |
| $0.1(\Gamma = 0.99)$ | 0.05 | $3.1 \times 10^{-3}$ | $4.5 \times 10^{-4}$ | **6.9** |
| $0.1(\Gamma = 0.95)$ | 0.05 | $1.1 \times 10^{-3}$ | $5.4 \times 10^{-4}$ | 2.0 |
| $0.1(\Gamma = 0.95)$ | 0.2 | $5.5 \times 10^{-3}$ | $1.6 \times 10^{-3}$ | 3.4 |
| $0.1(\Gamma = 0.95)$ | 0.4 | $3.5 \times 10^{-3}$ | $1.3 \times 10^{-3}$ | 2.6 |
| $0.1(\Gamma = 0.95)$ | 0.5 | $8.0 \times 10^{-3}$ | $2.4 \times 10^{-3}$ | 3.3 |
| $0.11(\Gamma = 0.95)$ | 0.4 | $5.0 \times 10^{-3}$ | $1.1 \times 10^{-3}$ | 4.4 |

and LSN. Notably, the parameters $r = 0.11$, $\sigma = 0.4$, the test error magnitude of PINNs and reaches $10^{-3}$ and $10^{-3}$, respectively.

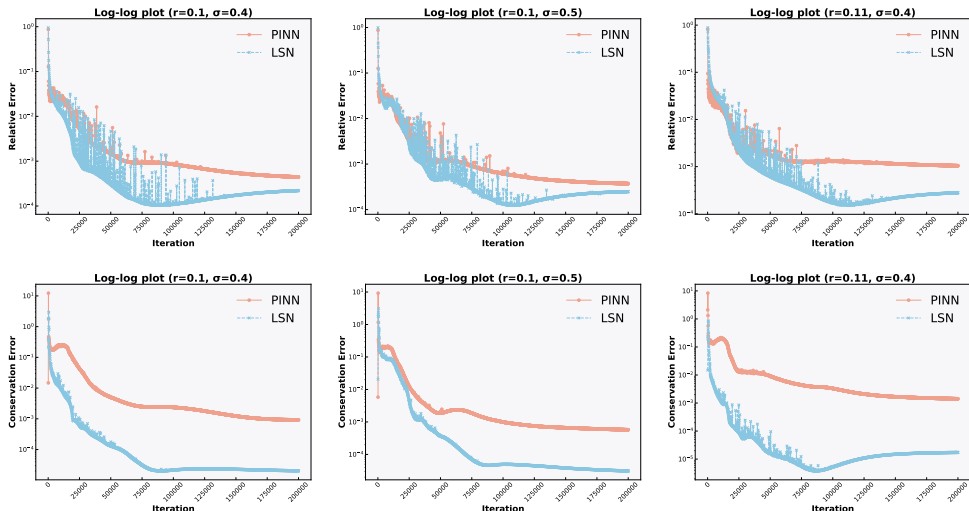

Figure 4: Log-log relative test error curves of PINNs and LSN under the forth parameters configuration.

Table 3: Comparison of cumulative training time for LSN and PINNs to achieve different error thresholds.

| Error Threshold | Cumulative Training Time (s) | |
| --- | --- | --- |
| | LSN | PINN |
| $7.00 \times 10^{-3}$ | $8.85 \times 10^{2}$ | $1.99 \times 10^{3}$ |
| $6.00 \times 10^{-3}$ | $8.92 \times 10^{2}$ | $2.26 \times 10^{3}$ |
| $5.00 \times 10^{-3}$ | $9.99 \times 10^{2}$ | $2.60 \times 10^{3}$ |
| $4.00 \times 10^{-3}$ | $1.11 \times 10^{3}$ | $3.25 \times 10^{3}$ |
| $3.00 \times 10^{-3}$ | $1.35 \times 10^{3}$ | $4.44 \times 10^{3}$ |

To provide a more intuitive demonstration of the superiority of LSN, we consider the test accuracy of the method under different equation parameters, as shown in Table 2. Compared with vanilla PINNs, LSN can reduce the relative test error by up to **7** times, with an average improvement of **2-4** times.

In addition, we conducted additional experiments on training efficiency under the parameter settings $r = 0.2$ and $\sigma = 0.2$. In these experiments, we compared the cumulative training time required for LSN and PINNs to achieve specified error thresholds. The results are summarized in Table 3. As observed, LSN exhibited a consistent and significant efficiency advantage across all tested error thresholds. For instance, at an error threshold of $7 \times 10^{-3}$, LSN achieved the target precision in $8.85 \times 10^{2}$ seconds, whereas PINNs required $1.99 \times 10^{3}$ seconds. Despite the addition of regularization terms in LSN, which complicates the loss function to be optimized, the experimental results indicate that LSN does not converge more slowly than PINNs in terms of wall-clock time and simultaneously achieves higher accuracy.

### 5.1.2 Comparison with State-of-the-art Methods

We conduct comparative experiments between LSN and several state-of-the-art methods including IPINNs (Bai et al., 2022), sfPINNs (Wong et al., 2022), ffPINNs (Wong et al., 2022) and LPS (Akhound-Sadegh et al., 2024), under different equation parameter setups (i.e., different risk-free rate and volatility), following Ankudinova & Ehrhardt (2008); Shinde & Takale (2012); Bai et al. (2022).

**Experimental Design.** All methods share the following hyperparameter setup, i.e., learning rate $lr = 0.001$ and learning rate decay rate $\Gamma = 0.95$. The training steps are set as 80,000 and 200,000, depending on the speed of convergence.

Figure 5 provides Log-log relative test error curves and function approximation results of LSN, PINNs, and PINNs variants. The results demonstrate that the relative error of LSN consistently remains below those of vanilla PINNs and their variants across different experimental settings. Both sfPINNs and ffPINNs

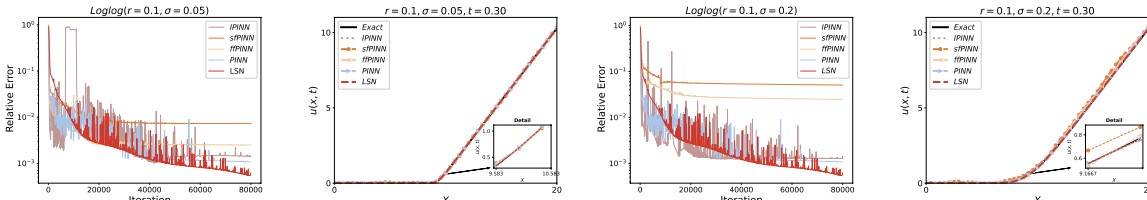

Figure 5: Log-log relative test error curves and function approximation results of LSN, PINNs, and PINNs variants. The error curves are shown in the first and third rows, while the function approximation results at $t = 0.30$ are presented in the second and fourth rows.

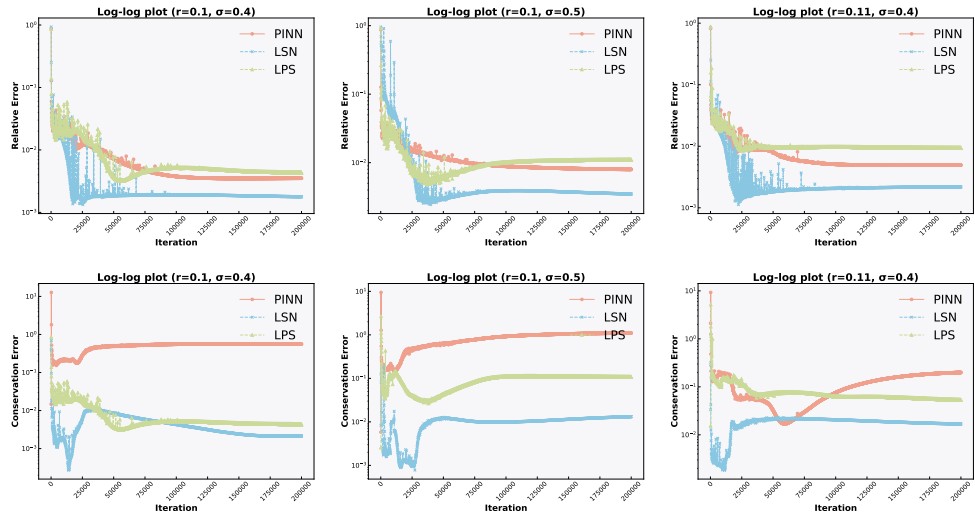

Figure 6: Log-log relative test error curves of PINNs, LSN and LPS under the third parameters configuration.

exhibit unsatisfactory performance under certain parameters, occasionally performing even worse than vanilla PINNs. This underperformance might be attributed to the fact that sfPINNs and ffPINNs are more suited to scenarios with sinusoidal-form solutions, thereby failing to effectively approximate the complex solution of the Black-Scholes equation (Wong et al., 2022).

We conduct comparative experiments among LSN, LPS, and PINNs using the same weights, as shown in Figure 6. Specifically, LSN and LPS share the same weights $\lambda_i$ for $i = 1, \ldots, 4$, while PINNs share the same weights $\lambda_i$ for $i = 1, \ldots, 3$ as LSN and LPS but with $\lambda_4 = 0$. The experiments demonstrate that LSN outperforms both PINNs and LPS. Additionally, LPS exhibits overall superior performance compared to PINNs when early stopping is employed.

For a more fine-grained comparison between LPS and LSN, we further finetune the weights $l_i$ $(i = 1, \ldots, 4)$ of the loss function of LPS under different configurations. Notably, $l_i$ $(i = 1, \ldots, 3)$ in LPS serve the same purpose as $\lambda_i$ $(i = 1, \ldots, 3)$ in LSN, while $l_4$ in LPS determines the weight of the symmetry residuals, and $\lambda_4$ in LSN determines the weight of the residuals of the conservation laws corresponding to the Lie symmetry operators. To illustrate the specific process of weight tuning for LPS, consider the example with $r = 0.1$ and $\sigma = 0.4$, as shown in Figure 7. We start by fixing all weights to 1 and then traverse $l_1$ values from $[10, 1, 0.1, 0.01, 0.001, 0.0001]$ in descending order, selecting the best value of $l_1 = 1$. Similarly, we traverse $l_2$ values and find that $l_2$ performs well in the range of 0.1 to 10. We then further subdivide this range into $[10, 4, 2, 1, 0.5, 0.25, 0.1]$ for experimentation and select the best value for $l_2$, which is fixed thereafter. This process is repeated for finetuning other parameters of LPS.

After finetuning the weights of the loss function of LPS, and using the previously set weights for LSN and PINNs, we validate the performance of LSN, PINNs, and the finetuned LPS model. As shown in Figure 8, after extensive weight tuning, LPS outperforms PINNs with early stopping in relative test error but remains surpassed by LSN in terms of relative test error and conservation law error.

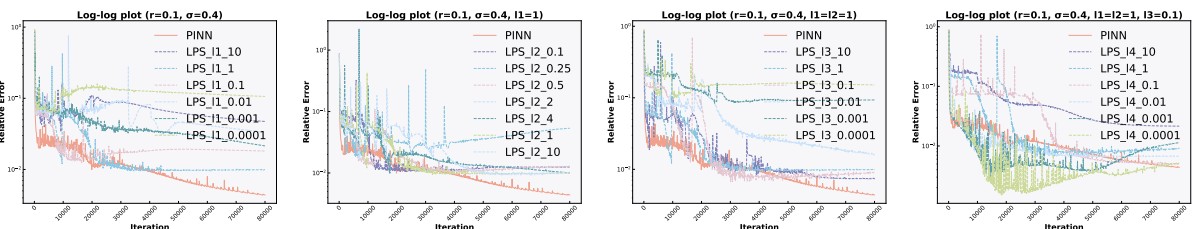

Figure 7: Log-log relative test error curves of PINNs and LPS.

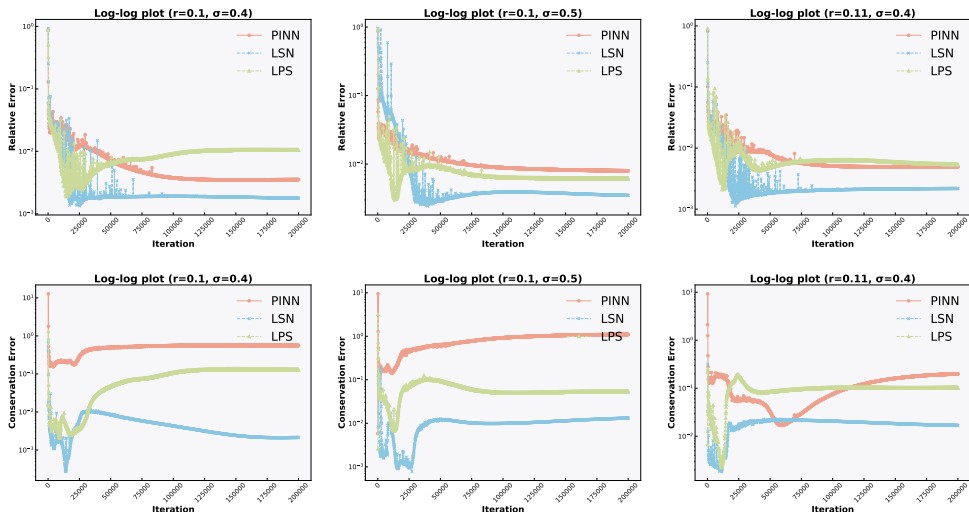

Figure 8: Log-log Error Curves Over Training Steps: PINNs vs. LSN vs. LPS. After individually tuning the weights for LPS, the performance of LSN is evaluated on the testing collocation data set using two metrics.

This observation can be explained by the fact that LSN potentially retains a broader spectrum of symmetries than LPS by leveraging conservation law systems. These systems incorporate additional symmetries, including potential symmetries (Bluman, 1993; Pucci & Saccomandi, 1993), nonlocal symmetries (Bluman et al., 1988; Olver, 1993), and novel local symmetries (Edelstein & Govinder, 2009). By utilizing the system of conservation laws, the symmetry analysis of the original equation is extended, thus revealing previously unexplored structures and solutions.

We validate this claim with the following experiments. For the equation parameters $r = 0.1$ and $\sigma = 0.4$ as shown in Figure 8, we conducted experiments with 100k training steps. The Lie symmetry operator $G_2$ is unique, while the derived conservation laws are not, as they involve freely chosen functions $l(t)$ and $g(t)$ (see Equation (7)). We defined two conservation law operators: LSN: $O_1$ with $l(t) = t$ and $g(t) = t^2$, and LSN: $O_2$ with $l(t) = \sin(t)$ and $g(t) = \cos(t)$. We also combined these operators as LSN: $O_1 + O_2$. As shown in Figure 9, upon convergence, both LSN: $O_1$ and LSN: $O_2$ demonstrate superior performance compared to PINN and LPS, with the combined operator LSN: $O_1 + O_2$ achieving the highest accuracy. These results indicate that the flexibility in choosing $l(t)$ and $g(t)$ allows the conservation laws to integrate additional information, thereby enhancing solution accuracy.

## 5.2 Experiments on Other Differential Equations

This section illustrates the methodological flexibility of LSN when applied to the Vašiček model, the Maxwellian tails model, and the KdV model. In particular, experimental results on the Vašiček model with different operator combinations indicate that stacking multiple operators can potentially improve accuracy.

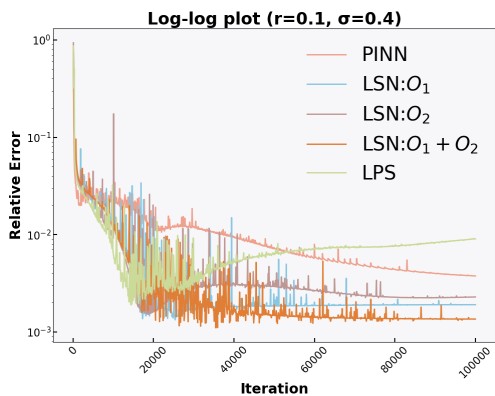

Figure 9: Log-log Error Curves Over Training Steps: PINNs vs. LSN vs. LPS. For $O_1$, the functions are defined as $l(t) = t$ and $g(t) = t^2$ (see Eq. (7)). For $O_2$, $l(t) = \sin(t)$ and $g(t) = \cos(t)$. The resulting operator $O_1 + O_2$ represents the combined effect through linear superposition.

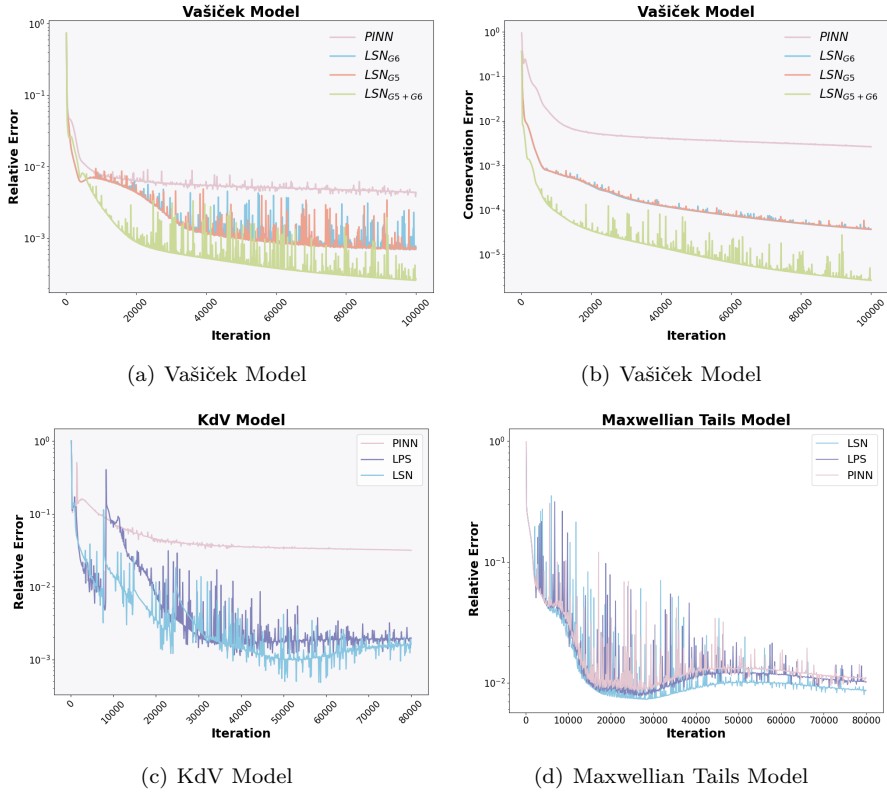

(a) Vašiček Model

(b) Vašiček Model

(c) KdV Model

(d) Maxwellian Tails Model

Figure 10: Log-log Error Curves Over Training Steps: PINNs vs. LSN vs. LPS. The first row denotes: For the Vašiček equation, $G_6$ $(G_5)$ denotes the use of a single symmetry operator corresponding to the conservation law, while $G_5 + G_6$ represents the linear combination of two operators. The second row denotes: The algorithms are further extended to the KdV model and the Maxwellian model.

### 5.2.1 Vašiček Equation

This section extends the LSN algorithm to the Vašiček equation (Privault, 2022), where experiments are performed using both individual operators and stacked combinations. A detailed description of the Vašiček model is provided in Appendix A.2.

**Experimental Design.** In this experiment, the parameters of the Vašiček model are set as follows: $\alpha = 0.03$, $\beta = 0.08$, $\gamma = -1$, $\sigma = 0.03$, $\Omega = 1$, and $T = 1$. The dataset consists of 500 interior points and 200 boundary

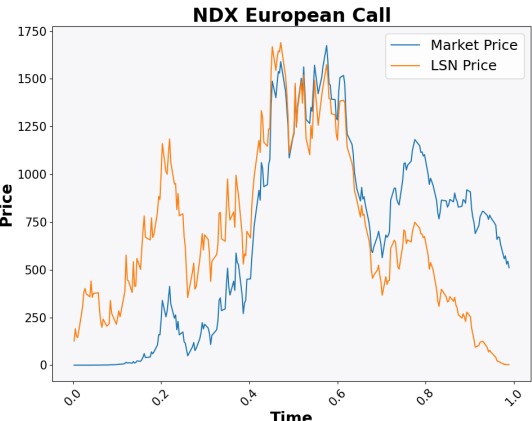

Figure 11: Predicting European call options on Nasdaq 100 index using real financial data with LSN.

points. The network is designed with two layers, each containing 10 neurons. The training is performed for 100,000 iterations with a learning rate of $lr = 0.001$ and a learning rate decay factor of $\Gamma = 0.95$.

As shown in Figure 10(a) and 10(b), extending LSN to the Vašiček equation further illustrates its ability to handle diverse operator structures. It is observed that while the use of a single operator already achieves significant improvements over PINNs, the performance gains achieved through combined operators are even more substantial. This demonstrates the flexibility of our method: it can effectively utilize both single operators and combinations of multiple operators.

### 5.2.2 KdV Model

This section applies the LSN algorithm to the KdV equation (Ibragimov, 2007), with experiments conducted using a single operator. The KdV equation, a nonlinear partial differential equation commonly used in fluid mechanics and related fields, is expressed as $u_t = u_{xxx} + uu_x$.

**Experimental Design.** The KdV model parameters are configured as follows: the spatial domain is $\Omega = [0, 1]$, and the temporal domain is $T = 1$. The dataset includes 100 interior points and 400 boundary points. The neural network architecture consists of 4 layers, each containing 50 neurons. Training is conducted for 200,000 iterations with a learning rate of $lr = 0.001$ and a learning rate decay factor of $\Gamma = 0.95$.

In the experiment using a single operator (see Figure 10(c)), LSN achieves a relative test error on the order of $10^{-4}$. This result is significantly better than the PINNs method (accuracy on the order of $10^{-2}$) and the LPS method (accuracy on the order of $10^{-3}$), demonstrating LSN's superior accuracy and stability in solving the KdV equation.

### 5.2.3 Maxwellian Tails Model

This section applies the LSN algorithm to the Maxwellian tails model. The Maxwellian tails model is used to describe the behavior of particles in a system where the distribution of velocities follows a Maxwellian distribution, with particular focus on the high-energy tails of this distribution, which is expressed as: $u_{xt} + u_x + u^2 = 0$.

**Experimental Design.** In this experiment, the parameters of the Maxwellian tails model are set as follows: the spatial domain is $\Omega = [1, 2]$, and the temporal domain is $T = 1$. The dataset includes 100 interior points and 100 boundary points. The neural network architecture is designed with 4 layers, each containing 50 neurons. Training is performed for 200,000 iterations with a learning rate of $lr = 0.001$ and a learning rate decay factor of $\Gamma = 0.95$.

In the experiment (as shown in Figure 10(d)), the decreasing relative error curve over training steps shows that LSN consistently outperforms LPS and PINNs. This result demonstrates the superior performance of LSN in the Maxwellian tails model.

### 5.3 Experiments on Real Market Data

In this section, we extend the experiments to real-time market data. The focus is on European call options based on the Nasdaq 100 index. The neural network used for option pricing relies on two fixed inputs: volatility ($\sigma$) and risk-free interest rate ($r$). In the experimental setup, the implied volatility from the OptionMetrics dataset (Wachowicz, 2020) is used. For the risk-free interest rate, the average yield of one-year Treasury bonds during the option period is taken as the average yield of one-year Treasury bonds during the option period, obtained by scraping data from Yahoo Finance. We train a LSN model for this type of option and defines the required range of spot prices. During inference, these independent models are used to predict market prices.

As shown in Figure 11, LSN performs well (Dhiman & Hu, 2023), but some discrepancies remain compared to market prices, potentially attributable to market supply and demand dynamics or other external influences. However, overall, the LSN method performs well, and a high correlation is observed between market prices and predicted prices.

## 6  Conclusion

This paper proposes Lie Symmetry Net (LSN) to solve differential equations for modeling financial market dynamics by exploiting the intrinsic symmetry of collocation data. The Lie symmetries of these equations is interpreted as several conservation laws. We define a Lie symmetries residual to measure how well these conservation laws are realised at specific points in the collocation data space, which is then integrated over the entire collocation data space to form a Lie symmetry risk. LSN is optimized under structural risk minimization framework to balance the Lie symmetry risk and the original collocation data fitting residuals. Extensive experiments demonstrate the effectiveness and scalability of LSN, showing that the test error is reduced by more than an order of magnitude.

### Broader Impact Statement and Future Work

This paper aims to develop AI-driven, symmetry-aware differential equation simulators to model financial market dynamics, which may also contribute to scientific discovery and engineering. This paper also pioneers the realization of Lie symmetries by maintaining the corresponding conservation laws, presenting a universal, off-the-shelf solution that is not limited to PINNs or the Black-Scholes equation, but can be extended to a wide range of backbones and differential equations. For future work, we will consider the incorporation of symmetries into network architecture.

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

# A   Appendix

The Appendix is divided into three parts: 1) Section A.1 provides the necessary definitions and lemmas, 2) Section A.2 includes the general form of the PDE, 3) Section A.3 provides the knowledge about the Lie symmetries of Vašiček, 4) Section 5.3 provieds the some experiments with different Lie symmetry operator conbinations, and 5) Section A.4 presents the theoretical analysis of LSN, including its approximation and generalization properties.

## A.1   Definitions and Technical Lemmas

In this section, we will present the definitions and lemmas required for our subsequent discussions.

**Definition A.1** (Wiener process)**.** The Wiener process (also known as Brownian motion) is a continuous-time stochastic process commonly used to model random walks. The standard definition of a Wiener process includes several key features:

1. Starting Point: The process starts at $W_0 = 0$, indicating that its initial position is zero.

2. Independent Increments: For all $0 \leq s < t$, the increments $W_t - W_s$ are mutually independent. This implies that the process is memory-less, and its future behavior is not influenced by its past.

3. Stationary Increments: For all $0 \leq s < t$, the distribution of the increment $W_t - W_s$ depends only on the time difference $t - s$, and is independent of the specific values of $s$ and $t$. Mathematically, this is expressed as $W_t - W_s \sim \mathcal{N}(0, t-s)$, where $\mathcal{N}(0, t-s)$ denotes a normal distribution with mean 0 and variance $t-s$.

4. Continuous Paths: The paths of the Wiener process are almost surely continuous. This means that the function $t \mapsto W_t$ is continuous with probability 1 .

**Definition A.2** (European call options)**.** European call options are financial derivatives granting the holder the right, without obligation, to purchase the underlying asset at a predetermined price upon expiration.

**Definition A.3** (Lie symmetries (Gazizov & Ibragimov, 1998))**.** Consider second-order evolutionary PDEs:

$$u_t - F(t, x, u, u_{(1)}, u_{(2)}) = 0, \tag{13}$$

where $u$ is a function of independent variables $t$ and $x = (x^1, \cdots, x^n)$, and $u_{(1)}$, $u_{(2)}$ represent the sets of its first and second-order partial derivatives: $u_{(1)} = (u_{x^1}, \cdots, u_{x^n})$, $u_{(2)} = (u_{x^1 x^1}, u_{x^1 x^2}, \cdots, u_{x^n x^n})$. Transformations of the variables $t$, $x$, $u$ are given by:

$$\bar{t} = f(t, x, u, a), \quad \bar{x}^i = g^i(t, x, u, a), \quad \bar{u} = h(t, x, u, a), \quad i = 1, \ldots, n, \tag{14}$$

where these transformations depend on a continuous parameter $a$. These are defined as symmetry transformations of Equation (13) if the equation retains its form in the new variables $\bar{t}, \bar{x}, \bar{u}$. The collection $G$ of all such transformations forms a continuous group, meaning $G$ includes the identity transformation:

$$\bar{t} = t, \quad \bar{x}^i = x^i, \quad \bar{u} = u,$$

the inverse of any transformation in $G$, and the composition of any two transformations in $G$. This symmetry group $G$ is also known as the group admitted by Equation (13). According to the Lie group theory, constructing the symmetry group $G$ is equivalent to determining its infinitesimal transformations:

$$\bar{t} \approx t + a\xi^0(t, x, u), \quad \bar{x}^i \approx x^i + a\xi^i(t, x, u), \quad \bar{u} \approx u + a\eta(t, x, u). \tag{15}$$

For convenience, the infinitesimal transformation Equation (15) can be represented by the operator:

$$X = \xi^0(t, x, u)\frac{\partial}{\partial t} + \xi^i(t, x, u)\frac{\partial}{\partial x^i} + \eta(t, x, u)\frac{\partial}{\partial u}.$$

**Definition A.4** (Conservation law (Khalique & Motsepa, 2018))**.** Any Lie point, Lie–Bäcklund or non-local symmetry

$$X = \xi^i(x, u, u^{(1)}, \ldots)\frac{\partial}{\partial x^i} + \eta(x, u, u^{(1)}, \ldots)\frac{\partial}{\partial u},$$

of differential equation

$$F(x, u, u^{(1)}, \ldots, u^{(s)}) = 0 \tag{16}$$

provides a conservation law

$$D_i(T^i) = 0,$$

for the system of differential equations comprising Equation (16) and the adjoint equation

$$F^*(x, u, v, u^{(1)}, v^{(1)}, \ldots, u^{(s)}, v^{(s)}) = \frac{\delta(vF)}{\delta u}.$$

The conserved vector is given by

$$
\begin{aligned}
T^i &= \xi^i L + W\left[\frac{\partial L}{\partial u_i} - D_j\left(\frac{\partial L}{\partial u_{ij}}\right) + D_j D_k\left(\frac{\partial L}{\partial u_{ijk}}\right) - \cdots\right] \\
&\quad + D_j(W)\left[\frac{\partial L}{\partial u_{ij}} - D_k\left(\frac{\partial L}{\partial u_{ijk}}\right) + \cdots\right] + D_j D_k(W)\left[\frac{\partial L}{\partial u_{ijk}} - \cdots\right] + \cdots,
\end{aligned}
$$

where $W$ and $L$ are defined as

$$W = \eta - \xi^j u_j, \quad L = vF(x, u, u^{(1)}, \ldots, u^{(s)}).$$

**Definition A.5** (Relative test error)**.** The relative test error between an approximate solution $\hat{u}(\mathcal{S})$ and an exact solution $u^*(\mathcal{S})$ on test collocation data $\mathcal{S}$ is defined as follows:

$$\text{Relative test error} = \left\|\frac{\hat{u}(\mathcal{S}) - u^*(\mathcal{S})}{u^*(\mathcal{S})}\right\|.$$

**Definition A.6.** [Feynman-Kac formula](Del Moral & Del Moral, 2004) The Feynman-Kac formula provides a critical theoretical framework to establish a connection between certain types of PDEs and SDEs. Given a payoff function $f(x, t)$ and defined a discounting function $r(x, t)$ to calculate the present value of future payoffs, if $u(x, t)$ is a solution to the PDE:

$$\frac{\partial u}{\partial t} + \mu(x, t)\frac{\partial u}{\partial x} + \frac{1}{2}\sigma^2(x, t)\frac{\partial^2 u}{\partial x^2} - r(x, t)u = 0, \tag{17}$$

with the terminal condition $u(x, T) = f(x)$, then the solution $u(x, T)$ to this PDE can be represented as:

$$E\left[e^{-\int_t^T r(X_s, s)ds}f(X_T) \mid X_t = x\right], \tag{18}$$

where $X_T$ denotes the value of Equation (2) at time $T$.

**Lemma A.7** (Itô's lemma (Hassler, 2016))**.** *The general form of Itô's Lemma for a function $f(t, X_t)$ of time $t$ and a stochastic process $X_t$ satisfying a stochastic differential equation is given by*

$$df(t, X_t) = \left(\frac{\partial f}{\partial t} + \mu\frac{\partial f}{\partial x} + \frac{1}{2}\sigma^2\frac{\partial^2 f}{\partial x^2}\right)dt + \sigma\frac{\partial f}{\partial x}dW_t.$$

*Here $t$ represents time, $X_t$ is a stochastic process satisfying a stochastic differential equation $dX_t = \mu dt + \sigma dW_t$, $f(t, X_t)$ is the function of interest. $\frac{\partial f}{\partial t}, \frac{\partial f}{\partial x}$, and $\frac{\partial^2 f}{\partial x^2}$ denote the partial derivatives of $f$ with respect to time and the state variable $x$, $\mu$ is the drift coefficient in the SDE, $\sigma$ is the diffusion coefficient in the SDE and $dW_t$ is the differential of a Wiener process.*

**Lemma A.8** (Dynkin's formula (Klebaner, 2012)). *For every $x \in \Omega$, let $X^x$ be the solution to a linear PDE Equation (1) with affine $\mu : \mathbb{R}^d \to \mathbb{R}^d$ and $\sigma : \mathbb{R}^d \to \mathbb{R}^{d \times d}$. If $\varphi \in C^2(\mathbb{R}^d)$ with bounded first partial derivatives, then it holds that $(\partial_t u)(x,t) = \mathcal{L}[u](x,t)$ where $u$ is defined as*

$$u(x,t) = \varphi(x) + \mathbb{E}\left[\int_0^t (\mathcal{F}\varphi)(X_\tau^x)\, d\tau\right], \quad \text{for } x \in \Omega, \quad t \in [0, T], \tag{19}$$

*where*

$$dX_t^x = \mu(X_t^x)dt + \sigma(X_t^x)dW_t, \quad X_0^x = x,$$

$$(\mathcal{F}\varphi)(X_t^x) = \sum_{i=1}^d \mu_i(X_t^x)(\partial_i \varphi)(X_t^x) + \frac{1}{2}\sum_{i,j,k=1}^d \sigma_{i,k}(X_t^x)\sigma_{kj}(X_t^x)(\partial_{ij}^2 \varphi)(X_t^x), \tag{20}$$

*where $W_t$ is a standard d-dimensional Brownian motion on probability space $(\Omega, \mathcal{F}, P, (\mathbb{F}_t)_{t \in [0,T]})$, and $\mathcal{F}$ is the generator of $X_t^x$.*

**Lemma A.9** ((De Ryck & Mishra, 2022)). *Let $d, L, W \in \mathbb{N}, R \geq 1, L, W \geq 2$, let $\mu$ be a probability measure on $\Omega = [0,1]^d$, let $f : \Omega \to [-R(W+1), R(W+1)]$ be a function and let $f_\theta : \Omega \to \mathbb{R}, \theta \in \Theta$, be tanh neural networks with at most $L - 1$ hidden layers, width at most $W$ and weights and biases bounded by $R$. For every $0 < \epsilon < 1$, it holds for the generalisation and training error Equation (12) that,*

$$\mathbb{P}\left(\mathcal{E}_G(\theta^*(\mathcal{S})) \leq \epsilon + \mathcal{E}_T(\theta^*(\mathcal{S}), \mathcal{S})\right) \geq 1 - \eta \quad \text{if} \quad N \geq \frac{64d(L+3)^2 W^6 R^4}{\epsilon^4} \ln\left(\frac{4\sqrt[5]{d+4}RW}{\epsilon}\right).$$

## A.2 General PDEs

In this section, we will demonstrate the transformation of the BS equation and the Vašiček equation into a general form, i.e., Equation (1).

**Black-Scholes equation.** As detailed in the main text, the specific expression of the Black-Scholes Equation (4) is

$$\begin{cases} \frac{\partial u_\prime}{\partial t_\prime} + \frac{1}{2}\sigma^2 x_\prime^2 \frac{\partial^2 u_\prime}{\partial x_\prime^2} + rx_\prime \frac{\partial u_\prime}{\partial x_\prime} - ru_\prime = 0, & (x_\prime, t_\prime) \in \Omega \times [0, T], \\ u_\prime(T, x_\prime) = \max(x_\prime - K, 0), & x_\prime \in \Omega, \\ u_\prime(t_\prime, 0) = 0, & t_\prime \in [0, T]. \end{cases} \tag{21}$$

Let's $t = T - t_\prime, \in [T, 0], x = x_\prime \in \Omega$. Then the BS Equation (21) can be transformed into a more generalised initial-boundary value problem (Cervera, 2019),

$$\begin{cases} -\frac{\partial u}{\partial t} + \frac{1}{2}\sigma^2 x^2 \frac{\partial^2 u}{\partial x^2} + rx \frac{\partial u}{\partial x} - ru = 0, & (x, t) \in \Omega \times [0, T], \\ u(0, x) = \max(x - K, 0), & x \in \Omega \\ V(t, 0) = 0, & t \in [T, 0]. \end{cases} \tag{22}$$

Here $\mathbf{L}[u]$ in Equation (1) for Equation (22) is $\mathbf{L}[u] = \frac{1}{2}\sigma^2 x^2 \frac{\partial^2 u(x,t)}{\partial x^2} + rx \frac{\partial u(x,t)}{\partial x} - ru(x,t)$ with $\sigma(x) = \sigma x$, $\mu(x) = rx$, $\upsilon(x) = -r$ and $\varphi(x) = max(x - K, 0)$.

**Vašiček Equation (Privault, 2022).** In a financial market characterised by short-term lending transactions between financial institutions, the evolution of short-term interest rates can be modeled by the following SDE:

$$dx_t = \lambda(\beta - x_t)dt + \sigma dW_t,$$

where $\lambda, \beta > 0$, $\sigma$ are constants and $W_t$ is the Wiener process. Using the Feynman-Kac formula we can obtain the Vašiček pricing equation which is used to price risk-free bonds $u(x, t)$:

$$\begin{cases} \frac{\partial u}{\partial t} + \alpha \frac{\partial u}{\partial x^2} + \lambda(\beta - x)\frac{\partial u}{\partial x} + \gamma xu = 0 & \Omega \times [0, T], \\ u(x, T) = 1 & \Omega \times T, \\ u(x, t) = \psi(x, t) & \partial\Omega \times [0, T]. \end{cases} \tag{23}$$

where $\alpha = \frac{1}{2}\sigma^2$, $\gamma = -1$ and $\psi(x,t)$ is the boundary conditions. The zero-coupon bond price in the Vašiček pricing model is given by (Khalique & Motsepa, 2018):

$$u(x,t) = e^{A(T-t)+xC(T-t)}, \tag{24}$$

where $C(t) = -\frac{1}{\lambda}\left(1 - e^{-\lambda t}\right)$ and $A(t) = \frac{4\lambda^2\beta - 3\sigma^2}{4\lambda^3} + \frac{\sigma^2 - 2\lambda^2\beta}{2\lambda^2}t + \frac{\sigma^2 - \lambda^2\beta}{\lambda^3}e^{-\lambda t} - \frac{\sigma^2}{4\lambda^3}e^{-2\lambda t}$.

Similarly, we can express the Vašiček equation in a general form as follows:

$$\begin{cases} u_t(x,t) = \mathcal{L}[u], & \text{for all } (x,t) \in \Omega \times [0,T], \\ u(0,x) = \varphi(x), & \text{for all } x \in \Omega, \\ u(y,t) = \psi(y,t), & \text{for all } (y,t) \in \partial\Omega \times [0,T], \end{cases} \tag{25}$$

where $\mathbf{L}[u]$ in Equation (1) for Equation (23) is $\mathbf{L}[u] = \alpha u_{xx} + \lambda(\beta - x)u_x + \gamma xu$ with $\sigma(x) = \sqrt{2\alpha}$, $\mu(x) = \lambda(\beta - sx)$, $\upsilon(x) = \gamma x$ and $\varphi(x) = 1$.

### A.3 Lie Symmetries of Vašiček Equation

**Lie Symmetry Operator.** Lie symmetry operator is a major mathematical tool for characterizing the symmetry in PDEs (see Definition A.3) (Paliathanasis et al., 2016). The Lie symmetry operators (Khalique & Motsepa, 2018) of Vašiček Equation (23) are given by the vector field

$$\begin{aligned} G_\phi &= \phi(t,x)\frac{\partial}{\partial u}, G_1 = \frac{\partial}{\partial t}, \\ G_2 &= e^{2\lambda t}\frac{\partial}{\partial t} + \frac{e^{2\lambda t}}{\lambda}\left(\lambda^2 x - 2\alpha\gamma - \beta\lambda^2\right)\frac{\partial}{\partial x} + \frac{ue^{2\lambda t}}{\alpha\lambda^2}\left(\alpha^2\gamma^2 + 2\alpha\beta\gamma\lambda^2 - \alpha\lambda^3 - 3\alpha\gamma\lambda^2 x + \lambda^4(\beta - x)^2\right)\frac{\partial}{\partial u}, \\ G_3 &= e^{-2\lambda t}\left[-\frac{\partial}{\partial t} + \frac{1}{\lambda}\left(\lambda^2(x - \beta) - 2\alpha\gamma\right)\frac{\partial}{\partial x} + \frac{\gamma u}{\lambda^2}\left(\lambda^2 x - \alpha\gamma\right)\frac{\partial}{\partial u}\right], \\ G_4 &= e^{\lambda t}\left[\frac{\partial}{\partial x} + \frac{u}{\alpha\lambda}\left(-\alpha\gamma - \beta\lambda^2 + \lambda^2 x\right)\frac{\partial}{\partial u}\right], G_5 = e^{-\lambda t}\left[\frac{\partial}{\partial x} + \frac{\gamma u}{\lambda}\frac{\partial}{\partial u}\right], G_6 = u\frac{\partial}{\partial u}. \end{aligned} \tag{26}$$

These Lie symmetry operators not only provide a deeper insight into the structure of the PDEs but also form the foundation for deriving conservation laws associated with these equations.

**Conservation Law.** To ascertain the Lie conservation law operators for the Vašiček Equation (23), it is necessary to analyze its adjoint equation, as follows

$$\frac{\partial\nu}{\partial t} - \alpha\frac{\partial\nu}{\partial x^2} - \lambda(x - \beta)\frac{\partial\nu}{\partial x} - (\lambda + \gamma x)\nu = 0. \tag{27}$$

where $\nu \neq 0$ is a new dependent variable $\nu = e^{pt+qx}$ with $p = \alpha q^2 - \lambda\beta q + \lambda$ and $q = -\frac{\gamma}{\lambda}$(Here, only one example is presented for illustration purposes, although there exist numerous solutions to this set of adjoint equation). For illustrative purposes, we choose the relatively simple operator $G_5$ and $G_6$ as examples of the Vašiček Equation (26) and provide the corresponding conserved quantities (Khalique & Motsepa, 2018):

$$\begin{cases} T_5^t(u,x,t) = \frac{1}{\gamma\lambda}e^{-\lambda t}\nu\left(\lambda\frac{\partial u}{\partial x} - \gamma u\right), \\ T_5^x(u,x,t) = \frac{1}{\gamma\lambda}e^{-\lambda t}\left\{\gamma u\left(\alpha\frac{\partial\nu}{\partial x} - \beta\lambda\nu\right) - \alpha\frac{\partial u}{\partial x}\left(\gamma\nu + \lambda\frac{\partial\nu}{\partial x}\right) - \lambda\frac{\partial u}{\partial t}\nu\right\}; \end{cases}$$

$$\begin{cases} T_6^t(u,x,t) = u\nu, \\ T_6^x(u,x,t) = \alpha\frac{\partial u}{\partial x}\nu - u\left\{\lambda(x - \beta)\nu + \alpha\frac{\partial\nu}{\partial x}\right\}. \end{cases}$$

In Section 5.3, we validate the general applicability of LSN by extending it to the Vašiček model and showing the adaptability of different Lie symmetry operators.

## A.4 Theoretical Analysis

Given the wide range of choices for Lie symmetry operators, we use the BS equation with the selected lie operator $G_2 = x\frac{\partial}{\partial x}$ of Equation (5) as an example to theoretically demonstrate the effectiveness of our method. The corresponding conservation law Equation (6) is as follows,

$$\mathcal{R}_{Lie}[\hat{u}] := D_t T_2^t(\hat{u}) + D_x T_2^x(\hat{u}),\tag{28}$$

where

$$\begin{cases} T_2^t(\hat{u}) = -\hat{u}_x l(t) + \dfrac{a}{x} + \dfrac{2b\hat{u}}{\sigma^2 x}\mathrm{e}^{-rt}, \\[2ex] T_2^x(\hat{u}) = \hat{u}_t l(t) + \hat{u}l'(t) + g(t) - b\hat{u}\mathrm{e}^{-rt} + b\left(\hat{u}_x + \dfrac{2r\hat{u}}{\sigma^2 x}\right)x\mathrm{e}^{-rt}. \end{cases}\tag{29}$$

Performing operator calculations with the conserved quantities $(T_2^t, T_2^x)$ substituted into the Equation (28) yields

$$\begin{cases} D_t T_2^t(\hat{u}) = -\hat{u}_{xt}l(t) - \hat{u}_x l_t(t) + \dfrac{2b\hat{u}_t}{\sigma^2 x}e^{-rt} - \dfrac{2rb\hat{u}}{\sigma^2 x}e^{-rt}, \\[2ex] D_x T_2^x(\hat{u}) = \hat{u}_{tx}l(t) + \hat{u}_x l_t(t) - b\hat{u}_x e^{-rt} + b\left(\hat{u}_{xx} + \dfrac{2r\hat{u}_x}{\sigma^2 x} - \dfrac{2r\hat{u}}{\sigma^2 x^2}\right)xe^{-rt} + b\left(\hat{u}_x + \dfrac{2r\hat{u}}{\sigma^2 x}\right)e^{-rt}. \end{cases}$$

Therefore, we have

$$\begin{aligned} D_t T_2^t(\hat{u}) + D_x T_2^x(\hat{u}) &= \frac{2b\hat{u}_t}{\sigma^2 x}e^{-rt} - b\hat{u}_x e^{-rt} + bx\hat{u}_{xx}e^{-rt} + \frac{2rb\hat{u}_x}{\sigma^2}e^{-rt} - \frac{2rb\hat{u}}{\sigma^2 x}e^{-rt} + b\hat{u}_x e^{-rt} \\ &= bxe^{-rt}\hat{u}_{xx} + \frac{2b}{\sigma^2 x}e^{-rt}\hat{u}_t + \frac{2rb}{\sigma^2}e^{-rt}\hat{u}_x - \frac{2rb}{\sigma^2 x}e^{-rt}\hat{u} \\ &= \frac{2be^{-rt}}{\sigma^2 x}\left(\hat{u}_t + \frac{1}{2}\sigma^2 x^2 \hat{u}_{xx} + rx\hat{u}_x - r\hat{u}\right). \end{aligned}$$

Since $b$ is arbitrarily chosen, let's set $b = x_{min}$. Where $x_{min}$ represents the smallest x-coordinate among the points in the configuration set. And $(x, t) \in \Omega \times [0, T]$ represents a bounded interior region, where $x$ and $t$ are within the specified domain $\Omega$ and time interval $[0, T]$ respectively. Therefore, there exists a positive number $M > 0$ such that

$$0 < \left|\frac{2be^{-rt}}{\sigma^2 x}\right|^2 < \left\|\frac{2be^{-rt}}{\sigma^2 x}\right\|_\infty^2 = \left\|\frac{2x_{min}e^{-rt}}{\sigma^2 x}\right\|_\infty^2 \le \left(\frac{2x_{min}e^{-rT}}{\sigma^2 x_{min}}\right)^2 \le \left(\frac{2e^{-rT}}{\sigma^2}\right)^2 := M.\tag{30}$$

Therefore,

$$\begin{aligned} \mathcal{L}_{Lie}[\hat{u}] = \|\mathcal{R}_{Lie}[\hat{u}]\|^2 &= \|D_t T_2^t(\hat{u}) + D_x T_2^x(\hat{u})\|^2 \\ &= \|\frac{2be^{-rt}}{\sigma^2 x}\left(\hat{u}_t + \frac{1}{2}\sigma^2 x^2 \hat{u}_{xx} + rx\hat{u}_x - r\hat{u}\right)\|^2 \\ &\le M\|\hat{u}_t + \frac{1}{2}\sigma^2 x^2 \hat{u}_{xx} + rx\hat{u}_x - r\hat{u}\|^2 \\ &= M\|\mathcal{R}_{PDE}[\hat{u}]\|^2. \end{aligned}\tag{31}$$

### A.4.1 Approximation Error Bounds of LSN

The PDE in Equation (1) is a linear parabolic equation with smooth coefficients, and conclusions about the existence of a unique classical solution $u$ to the equation, which is sufficiently regular, can be derived using standard parabolic theory. If $u$ is considered a classical solution, then the residual concerning $u$ should be zero.

$$\mathcal{R}_i[u](x, t) = 0, \quad \mathcal{R}_s[u](y, t) = 0, \quad \mathcal{R}_t[u](x) = 0, \quad \mathcal{R}_{Lie}[u](x, t) = 0, \quad \forall x \in \Omega, \quad y \in \partial\Omega.\tag{32}$$

Here $\mathcal{R}_{Lie}[u](x, t) = \frac{2be^{-rt}}{\sigma^2 x}\left(u_t + \frac{1}{2}\sigma^2 x^2 u_{xx} + rxu_x - ru\right) = \frac{2be^{-rt}}{\sigma^2 x}\mathcal{R}_i[u](x, t) = 0$ (with $\frac{2be^{-rt}}{\sigma^2 x} \neq 0$.) We first list several crucial lemmas used to prove the approximation error of LSN.

**Lemma A.10.** *Let $T > 0$ and $\gamma, d, s \in \mathbb{N}$ with $s \geq 2 + \gamma$. Suppose $u \in W^{s,\infty}\left((0,1)^d \times [0,T]\right)$ is the solution to a linear PDE equation 1. Then, for every $\varepsilon > 0$ there exists a tanh neural network $\widehat{u}^\varepsilon = u_{\widehat{\theta}^\varepsilon}$ with two hidden layers of width at most $\mathcal{O}\left(\varepsilon^{-d/(s-2-\gamma)}\right)$ such that $\mathcal{E}\left(\widehat{\theta}^\varepsilon\right) \leq \varepsilon$.*

*Proof.* We extend the proof of the Theorem 1 in De Ryck & Mishra (2022) to the LSN algorithm with regularization terms incorporating Lie symmetries. There exists a tanh neural network $\widehat{u}^\varepsilon$ with two hidden layers of width at most $\mathcal{O}\left(\varepsilon^{-d/(s-2-\gamma)}\right)$ such that

$$\|u - \widehat{u}^\varepsilon\|_{W^{2,\infty}((0,1)^d \times [0,T])} \leq \varepsilon.$$

Due to the linearity of PDEs (where Equation (1) is a linear equation with respect to $u$), it immediately follows that $|\mathcal{R}_i[u]|_{L^2((0,1)^d \times [0,T])} \leq \varepsilon$ and $|\mathcal{R}_{Lie}[u]|_{L^2((0,1)^d \times [0,T])} \leq M |\mathcal{R}_i[u]|_{L^2((0,1)^d \times [0,T])} \leq \varepsilon$. By employing a standard trace inequality, one can establish similar bounds for $\mathcal{R}_s[u]$ and $\mathcal{R}_t[u]$. Consequently, it directly follows that $\mathcal{E}\left(\widehat{\theta}^\varepsilon\right) \leq \varepsilon$. □

This lemma shows that the structure risk of LSN in Equation (12) can converge to zero. To address the challenge of the curse of dimensionality in structure risk of LSN Equation (12) bounds, we will leverage Dynkin's Lemma A.8, which establishes a connection between the linear partial differential Equation (1) and the Itô diffusion stochastic equation. Next, we will extend the proof for PINNs from De Ryck & Mishra (2022) to LSNs to demonstrate that the loss for LSNs can be made infinitesimally small.

**Lemma A.11.** *Let $\alpha, \beta, \varpi, \zeta, T > 0$, and $p > 2$. For any $d \in \mathbb{N}$, define $\Omega_d = [0,1]^d$ and consider $\varphi_d \in C^5\left(\mathbb{R}^d\right)$ with bounded first partial derivatives. Given the probability space $(\Omega_d \times [0,T], \mathcal{F}, \mu)$, and let $u_d \in C^{2,1}\left(\Omega_d \times [0,T]\right)$ be a function satisfying*

$$(\partial_t u_d)(x,t) = \mathbf{L}[u_d](x,t), \quad u_d(x,0) = \varphi_d(x), \quad \mathcal{L}_{Lie}[u_d](x,t) = 0 \quad \text{for all } (x,t) \in \Omega_d \times [0,T].$$

*Assume for every $\xi, \delta, c > 0$, there exist hyperbolic tangent (tanh) neural networks such that*

$$\|\varphi_d - \widehat{\varphi}_{\xi,d}\|_{C^2(D_d)} \leq \xi \quad \text{and} \quad \left\|\mathcal{F}\varphi - \widehat{(\mathcal{F}\varphi)_{\delta,d}}\right\|_{C^2([-c,c]^d)} \leq \delta. \tag{33}$$

*Under these conditions, there exist constants $C, \lambda > 0$ such that for every $\varepsilon > 0$ and $d \in \mathbb{N}$, a constant $\rho_d > 0$ (independent of $\varepsilon$) and a tanh neural network $\Psi_{\varepsilon,d}$ with at most $C\left(d\rho_d\right)^\lambda \varepsilon^{-\max\{5p+3, 2+p+\beta\}}$ neurons and weights that grow at most as $C\left(d\rho_d\right)^\lambda \varepsilon^{-\max\{\zeta, 8p+6\}}$ for $\varepsilon \to 0$ can be found such that*

$$\begin{aligned}
&\|\partial_t \Psi_{\varepsilon,d} - \mathbf{L}[\Psi_{\varepsilon,d}]\|_{L^2(\Omega_d \times [0,T])} + \|\Psi_{\varepsilon,d} - u_d\|_{H^1(\Omega_d \times [0,T])} \\
&+ \|\Psi_{\varepsilon,d} - u_d\|_{L^2(\partial(\Omega_d \times [0,T]))} + \|\mathcal{L}_{Lie}[\Psi_{\varepsilon,d}]\|_{L^2(\Omega_d \times [0,T])} \leq \varepsilon,
\end{aligned} \tag{34}$$

*where $\rho_d$ is defined as*

$$\rho_d := \max \sup_{\substack{x \in \Omega_d \\ s<t}} \sup_{s,t \in [0,T]} \frac{\|X_s^x - X_t^x\|_{\mathcal{L}^q\left(P, \|\cdot\|_{\mathbb{R}^d}\right)}}{|s-t|^{\frac{1}{p}}} < \infty. \tag{35}$$

*In this context, $X^x$ denotes the solution, following the Itô interpretation, of the stochastic differential equation (SDE) specified by Equation equation 13. Here $q > 2$ remains independent of $d$ and the norm $\|\cdot\|_{\mathcal{L}^q\left(P, \|\cdot\|_{\mathbb{R}^d}\right)}$ is defined as follows: Given a measure space $(\Omega, \mathcal{F}, \mu)$ where $q > 0$, for any $\mathcal{F}/\mathcal{B}(\mathbb{R}^d)$-measurable function $f : \Omega \to \mathbb{R}^d$,*

$$\|f\|_{\mathcal{L}^q(\mu, \|\cdot\|_{\mathbb{R}^d})} := \left[\int_\Omega \|f(\omega)\|_{\mathbb{R}^d}^q \mu(d\omega)\right]^{\frac{1}{q}}. \tag{36}$$

*Proof.* The main proof follows directly from Theorem 2 in De Ryck & Mishra (2022), where

$$\|\mathcal{L}_{Lie}[\Psi_{\varepsilon,d}]\|_{L^2(\Omega_d \times [0,T])} \leq M \|\partial_t \Psi_{\varepsilon,d} - \mathbf{L}[\Psi_{\varepsilon,d}]\|_{L^2(\Omega_d \times [0,T])} \leq \varepsilon. \tag{37}$$

□

According to Remark 2 by De Ryck & Mishra (2022), it is indicated that the assumption conditions in the Lemma A.11 are easily satisfied after modifications for the BS equation.

**Theorem A.12.** *Let $u$ be a classical solution to linear PDE as described in Equation (1) with $\mu \in C^1\left(\Omega; \mathbb{R}^d\right)$ and $\sigma \in C^2\left(\Omega; \mathbb{R}^{d \times d}\right)$, let $M = \left(\frac{2e^{-rT}}{\sigma^2}\right)^2$, $v \in C^2(\Omega \times [0,T]; \mathbb{R})$, and define the residuals according Equation (11). Then,*

$$\|u - v\|^2_{L^2(\Omega \times [0,T])} \leq C_1 \left[ \|\mathcal{R}_i[v]\|^2_{L^2(\Omega \times [0,T])} + \|\mathcal{R}_{lie}[v]\|^2_{L^2(\Omega \times [0,T])} + \|\mathcal{R}_t[v]\|^2_{L^2(\Omega)} \right.$$
$$\left. + C_2 \|\mathcal{R}_s[v]\|_{L^2(\partial\Omega \times [0,T])} + C_3 \|\mathcal{R}_s[v]\|^2_{L^2(\partial\Omega \times [0,T])} \right], \tag{38}$$

*where*

$$C_0 = 2 \sum_{i,j=1}^d \left\| \partial_{ij} \left(\sigma\sigma^T\right)_{ij} \right\|_{L^\infty(\Omega \times [0,T])},$$

$$C_1 = T e^{\left(2C_0 + 2\|\operatorname{div}\mu\|_\infty + 1 + \frac{1}{M} + 2\|v\|_\infty\right)T},$$

$$C_2 = 2 \sum_{i=1}^d \left\| \left(\sigma\sigma^T \nabla_x[u-v]\right)_i \right\|_{L^2(\partial\Omega \times [0,T])}, \tag{39}$$

$$C_3 = 2\|\mu\|_\infty + (1 + M) \sum_{i,j,k=1}^d \|\partial_i \left(\sigma_{ik}\sigma_{jk}\right)\|_{L^\infty(\partial\Omega \times [0,T])}.$$

*Proof.* Let $\hat{u} = v - u$. Integrating $\mathcal{R}_i[\hat{u}](t,x)$ over $\Omega$ and rearranging terms gives

$$\frac{d}{dt} \int_\Omega |\hat{u}|^2 dx = \frac{1}{2} \int_\Omega \operatorname{Trace}\left(\sigma^2 H_x[\hat{u}]\right) \hat{u} dx + \int_\Omega \mu J_x[\hat{u}]\hat{u} dx + \int_\Omega v|\hat{u}|^2 dx + \int_\Omega \mathcal{R}_i[\hat{u}]\hat{u} dx, \tag{40}$$

where all integrals are understood as integrals with respect to the Lebesgue measure on $\Omega$ and $\partial\Omega$, and where $J_x$ represents the Jacobian matrix, which is the transpose of the gradient with respect to the spatial coordinates. Following the derivation by Theorem 4 of De Ryck & Mishra (2022), we can similarly show that : for the first term

$$\int_\Omega \operatorname{Trace}\left(\sigma\sigma^T H_x[\hat{u}]\right) \hat{u} dx$$
$$\leq \sum_{i=1}^d \int_{\partial\Omega} \left|\left(\sigma\sigma^T J_x(\hat{u})^T\right)_i \hat{u} \left(\hat{e}_i \cdot \hat{n}\right)\right| dx - \underbrace{\int_\Omega J_x[\hat{u}]\sigma \left(J_x[\hat{u}]\sigma\right)^T dx}_{\geq 0} + \frac{c_2}{2} \int_{\partial\Omega} |\mathcal{R}_s[v]|^2 dx + \frac{c_3}{2} \int_\Omega \hat{u}^2 dx, \tag{41}$$

for the second term

$$\int_\Omega \mu J_x[\hat{u}]\hat{u} dx \leq \frac{1}{2}\|\operatorname{div}\mu\|_\infty \int_\Omega \hat{u}^2 dx + \frac{1}{2}\|\mu\|_\infty \int_{\partial\Omega} |\mathcal{R}_s[v]|^2 dx, \tag{42}$$

for the fourth term

$$\int_\Omega \mathcal{R}_i[\hat{u}]\hat{u} dx \leq \frac{1}{2} \int_\Omega \mathcal{R}_i[\hat{u}]^2 dx + \frac{1}{2} \int_\Omega \hat{u}^2 dx, \tag{43}$$

where $\hat{n}$ denotes the unit normal on $\partial\Omega$. $1 \leq i,j,k \leq d$ and

$$c_1 = 2 \sum_{i=1}^d \left\| \left(\sigma\sigma^T J_x[\hat{u}]^T\right)_i \right\|_{L^2(\partial\Omega \times [0,T])},$$
$$c_2 = \sum_{i,j,k=1}^d \|\partial_i \left(\sigma_{ik}\sigma_{jk}\right)\|_{L^\infty(\partial\Omega \times [0,T])}, \tag{44}$$
$$c_3 = \sum_{i,j=1}^d \left\| \partial_{ij} \left(\sigma\sigma^T\right)_{ij} \right\|_{L^\infty(\Omega \times [0,T])}.$$

As for the third term of Equation (40), we obtain

$$\int_\Omega v|\hat{u}|^2 dx \leq \|v\|_\infty \int_\Omega |\hat{u}|^2 dx. \tag{45}$$

Integrating Equation (40) over the interval $[0, \tau] \subset [0, T]$, using all the previous inequalities together with Hölder's inequality, we find that

$$
\begin{aligned}
\int_\Omega |\hat{u}(x,\tau)|^2 dx \leq{} & \int_\Omega |\mathcal{R}_t[v]|^2 \, dx + c_1 \left( \int_{\partial\Omega \times [0,T]} |\mathcal{R}_s[v]|^2 \, dxdt \right)^{1/2} + \int_{\Omega \times [0,T]} |\mathcal{R}_i[\hat{u}]|^2 \, dxdt \\
& + (c_2 + \|\mu\|_\infty) \int_{\partial\Omega \times [0,T]} |\mathcal{R}_s[v]|^2 \, dxdt + (c_3 + \| \operatorname{div} \mu\|_\infty + 1 + \|v\|_\infty) \int_{[0,\tau]} \int_\Omega |\hat{u}(x,s)|^2 dxdt.
\end{aligned}
\tag{46}
$$

Referring to Equation (40), we can transform operator $\mathcal{R}_{Lie}[\hat{u}] = \frac{2be^{-rt}}{\sigma^2 x} \left( u_t + \frac{1}{2}\sigma^2 x^2 u_{xx} + rxu_x - ru \right)$ with $\frac{2be^{-rt}}{\sigma^2 x} \neq 0$, i.e., $\mathcal{R}_i[\hat{u}] = u_t + \frac{1}{2}\sigma^2 x^2 u_{xx} + rxu_x - ru = \frac{\sigma^2 x}{2be^{-rt}}\mathcal{R}_{Lie}[\hat{u}]$ into the following form,

$$
\begin{aligned}
\frac{d}{dt} \int_\Omega |\hat{u}|^2 dx &= \frac{1}{2} \int_\Omega \operatorname{Trace} \left( \sigma^2 H_x[\hat{u}] \right) \hat{u} dx + \int_\Omega \mu J_x[\hat{u}]\hat{u} dx + \int_\Omega v|\hat{u}|^2 dx + \int_\Omega \frac{\sigma^2 x}{2be^{-rt}}\mathcal{R}_{Lie}[\hat{u}] dx \\
&\leq \frac{1}{2} \int_\Omega \operatorname{Trace} \left( \sigma^2 H_x[\hat{u}] \right) \hat{u} dx + \int_\Omega \mu J_x[\hat{u}]\hat{u} dx + \int_\Omega v|\hat{u}|^2 dx + \frac{1}{M} \int_\Omega \mathcal{R}_{Lie}[\hat{u}] dx.
\end{aligned}
\tag{47}
$$

The proof is ultimately established by using Using Grönwall's inequality and integrating over $[0, T]$.

$\square$

*Remark* A.13. Theorem A.12 states that by optimizing structure risk , the network's output can approximate the exact solution, while Lemma A.11 confirms that structure risk can be minimized. This verifies the numerical approximation of the LSN to the exact solution.

### A.4.2   Generalisation Error Bounds of LSN

We set a general configuration let $\Omega \subset \mathbb{R}^d$ be compact and let $u : \Omega \to \mathbb{R}, u_\theta : \Omega \to \mathbb{R}$ be functions for all $\theta \in \Theta$. We consider $u$ as the exact value of the PDE equation 1, and $u_\theta$ as the approximation generated by LSN with weights $\theta$.

Let $N \in \mathbb{N}$ be the training set size and let $\mathcal{S} = \{z_1, \ldots, z_M\} \in \Omega^N$ be the training set, where each $z_i$ is independently drawn according to some probability measure $\mu$ on $\Omega$. We define the structure risk and empirical loss as

$$
\mathcal{L}(\theta) = \int_\Omega |u_\theta(z) - u(z)|^2 \, d\mu(z), \quad \hat{\mathcal{L}}(\theta, \mathcal{S}) = \frac{1}{N} \sum_{i=1}^N |u_\theta(z_i) - u(z_i)|^2, \quad \theta^*(\mathcal{S}) \in \arg\min_{\theta \in \Theta} \hat{\mathcal{L}}(\theta, \mathcal{S}).
\tag{48}
$$

**Lemma A.14.** *Let $d, L, W \in \mathbb{N}$ with $R \geq 1$, and define $M = \left( \frac{2e^{-rT}}{\sigma^2} \right)^2$. Consider $u_\theta : [0,1]^d \to \mathbb{R}$, where $\theta \in \Theta$ representing tanh neural networks with at most $L-1$ hidden layers, each with a width of at most $W$, and weights and biases bounded by $R$. Let $\mathcal{L}^q$ and $\hat{\mathcal{L}}^q$ denote the structure risk and empirical error, respectively, for linear general PDEs as in Equation (1). Assume $\max\{\|\varphi\|_\infty, \|\psi\|_\infty\} \leq \max_{\theta \in \Theta} \|u_\theta\|_\infty$. Denote by $\mathfrak{L}^q$ the Lipschitz constant of $\mathcal{L}^q$, for $q = i, t, s$. Then, it follows that*

$$
\mathfrak{L}^q \leq 2^{5+2L} C (d+7)^2 L^4 R^{6L-1} W^{6L-6},
$$

*where $C = (1+M) \max\limits_{x \in D} \left( 1 + \sum\limits_{i=1}^d |\mu(x)_i| + \sum\limits_{i,j=1}^d \left| (\sigma(x)\sigma(x)^*)_{ij} \right| \right)^2$.*

*Proof.* Similar to Lemma 16 in De Ryck & Mishra (2022), we have the following:

$$
\begin{aligned}
\left| \mathcal{R}_i[u_\theta](t,x) - \mathcal{R}_i[\Phi^\vartheta](t,x) \right| &\leq |v(x)|_1 |u_\theta - \Psi^v|_\infty + (1 + |\mu(x)|_1) |J^\theta - J^\vartheta|_\infty + |\sigma(x)\sigma(x)^*|_1 |H_x^\theta - H_x^\vartheta|_\infty \\
&\leq 4\alpha (1 + |v(x)|_1 + |\mu(x)|_1 + |\sigma(x)\sigma(x)^*|_1)(d+7)L^2 R^{3L-1} W^{3L-3} 2^L |\theta - \vartheta|_\infty.
\end{aligned}
$$

And we have

$$
\left| \mathcal{R}_{lie}[u_\theta](t,x) - \mathcal{R}_{lie}[\Phi^\vartheta](t,x) \right| \leq M \left| \mathcal{R}_i[u_\theta](t,x) - \mathcal{R}_i[\Phi^\vartheta](t,x) \right|,
\tag{49}
$$

where we let $|\cdot|_p$ denote the vector $p$-norm of the vectored version of a general tensor. Next, we set $\vartheta = 0$ ) and $\max\{\|\varphi\|_\infty, \|\psi\|_\infty\} \leq \max_{\theta\in\Theta}\|u_\theta\|_\infty$ for $q = t, s$ that

$$
\begin{aligned}
\max_\theta \|\mathcal{R}_i[u_\theta]\|_\infty &\leq 4\alpha C_1(d+7)2^L L^2 R^{3L} W^{3L-3}, \\
\max_\theta \|\mathcal{R}_{lie}[u_\theta]\|_\infty &\leq 4\alpha C_1 M(d+7)2^L L^2 R^{3L} W^{3L-3}, \\
\max_\theta \|\mathcal{R}_q[u_\theta]\|_\infty &\leq 2WR,
\end{aligned}
\tag{50}
$$

where $C_1 = \max_{x\in\Omega}(1 + |v(x)|_1 + |\mu(x)|_1 + |\sigma(x)\sigma(x)^*|_1)$. Combining all the previous results yields the bound. $\qquad\square$

We can then obtain the generalization bound of LSN as follows.

**Theorem A.15.** *Let $L, W, N \in \mathbb{N}$, $R \geq 1$, $L, W \geq 2$, $a, b \in \mathbb{R}$ with $a < b$ and let $u_\theta : [0,1]^d \to \mathbb{R}$, $\theta \in \Theta$, be tanh neural networks with at most $L-1$ hidden layers, width at most $W$, and weights and biases bounded by $R$. For $q = i, t, s$, let $\mathcal{L}^q$ and $\hat{\mathcal{L}}^q$ denote the LSN structure risk and training error, respectively, for linear general PDEs as in Equation (1). Let $c_q > 0$ be such that $\hat{\mathcal{L}}^q(\theta, \mathcal{S})$, $\mathcal{L}^q(\theta) \in [0, c_q]$, for all $\theta \in \Theta$ and $S \subset \Omega^N$. Assume $\max\{\|\varphi\|_\infty, \|\psi\|_\infty\} \leq \max_{\theta\in\Theta}\|u_\theta\|_\infty$ and define the constants*

$$
C = (1+M)\max_{x\in\Omega}\left(1 + \sum_{i=1}^d |v(x)_i| + \sum_{i=1}^d |\mu(x)_i| + \sum_{i,j=1}^d \left|(\sigma(x)\sigma(x)^*)_{ij}\right|\right)^2.
$$

*Then, for any $\epsilon > 0$, it holds that*

$$
\mathcal{L}^q \leq \epsilon + \hat{\mathcal{L}}^q \quad \text{if } M_q \geq \frac{24dL^2W^2c_q^2}{\epsilon^4}\ln\left(4c_1RW\sqrt[6]{\frac{C(d+7)}{\epsilon^2}}\right).
\tag{51}
$$

*Proof.* The proof follows the generalization analysis of PINNs (De Ryck & Mishra, 2022). Setting

$$
C = (1+M)\max_{x\in D}\left(1 + \sum_{i=1}^d |v(x)_i| + \sum_{i=1}^d |\mu(x)_i| + \sum_{i,j=1}^d \left|(\sigma(x)\sigma(x)^*)_{ij}\right|\right)^2,
\tag{52}
$$

we can use Lemma A.14 with $a \leftarrow R, c \leftarrow c_q, \mathfrak{L} \leftarrow 2^{5+2L}C^2(d+7)^2L^4R^{6L-1}W^{6L-6}$ and $k \leftarrow 2dLW^2$ (Lemma A.9). We then arrive at

$$
k\ln\left(\frac{4a\mathfrak{L}}{\epsilon^2}\right) + \ln\left(\frac{2c_q}{\epsilon^2}\right) \leq 6kL\ln\left(4c_qRW\sqrt[6]{\frac{C(d+7)}{\epsilon^2}}\right) = 12dL^2W^2\ln\left(4c_qRW\sqrt[6]{\frac{C(d+7)}{\epsilon^2}}\right).
$$

$\qquad\square$

