# OpenReview forum: "Lie Symmetry Net: Preserving Conservation Laws in Modelling Financial Market Dynamics via  Differential Equations"
_TMLR — Accepted by TMLR_

### Review · Reviewer_KZfX · 2025-02-17

**Summary Of Contributions:**

The paper suggests an approach to include Lie symmetries into PINN loss functions for dealing with particual differential equations. The approaches are tested on SDEs from financial areas, which are transformed into PDEs via the Feynman-Kac formula. Superiority to some state of the art methods are shown on the Black-Scholes equation.

**Audience:**

Yes

**Claims And Evidence:**

Yes

**Requested Changes:**

Obviously, I would expect that all weaknesses are addressed in some form.

- p 2 l 6: I find it hard to follow to give the Lie symmetry without stating the Black-Scholes equation or the Lie group.
- p 2 l 7: "D_\cdot" might be mistaken for "D.". (See also Definition 4.1)
- p 4 l 4: "of partial" -> "of a partial"
- (1): This is not a general PDE. This is a "univariate evolution PDE with initial conditions and Dirichlet boundary conditions".
- p 5 l 5: "when groups are also differentiable manifolds, they are referred to as Lie groups" is wrong
- p 5 line after (5): the comma in "Z = r − σ2/2, ϕ(t, x)" makes it hard to parse this as two symbols.
- p 7: While it is pretty clear form the context of PINNs, one might add a (short) explanation how $\mathcal{R}_{Lie}$ is calculated
- p 7 l -9: What does "Gaussian sampling" mean here?
- The capitalization of the titles in the bibliography is off. While I grant that lower cases might be used, names should still be capitalized.

**Strengths And Weaknesses:**

Strengths
- The approach is relevant and the idea is sound.
- The paper structure is fine and I was mostly able to follow the paper.

Weaknesses
- The paper introduces a new method that should work for somewhat general PDEs with Lie symmetries. I find it disheartening to just test this method on a single PDE. This is a major drawback in this paper. (At least, it is made very clear, beginning from the abstract, that the paper takes this restriction.)
- There is no comparison
  - against neural operator approaches and
  - against state of the art approaches from numerical analysis.
 (PINN-based approaches have other advantages, perhaps you should stress them)
- No mention of computation times is given. Is the new approach at least comparable in computation time to previous approaches?
- I get that this is not a paper going deeper into SDEs. However, one should at least mention that applying the Feynman-Kac formula (e.g. form (3) to (4)) looses information. The deterministic PDE in (4) is just an expectation of the SDE in (3). This might seem like a minor point, but the paper claims (e.g. in the first line of the conclusion) that such SDE are being solved and not their expectations computed.
- The paper gives many citation (which is obviously a good idea), but is usually unprecise in its citation. For example the citation into (Rotman, 2012) on p5 cites a long book, and should better cite a specific chapter/page/definition. Or (Akhound-Sadegh et al., 2024) has the wrong year, see https://proceedings.neurips.cc/paper_files/paper/2023/hash/8493c860bec41705f7743d5764301b94-Abstract-Conference.html.
- While you cite state of the art methods, you do not describe differences. In particular, (Akhound-Sadegh et al., 2024) look very similar to your approach. There, I would like to see the differences be clearly described.

---

> ### Author Response · Authors · 2025-03-10
>
> >**Q1:**  The paper introduces a new method that should work for somewhat general PDEs with Lie symmetries. I find it disheartening to just test this method on a single PDE. This is a major drawback in this paper.
>
> **A1:**    Thanks for the suggestion. We conducted further experiments to include the Vasicek model ($\frac{\partial u}{\partial t} +\alpha \frac{\partial u}{\partial x^2}+\lambda(\beta-x)\frac{\partial u}{\partial x} +\gamma x u=0$) and other KDV 　equations($u_t=u_{xxx}+uu_x$). LSN consistently outperforms others in terms of test error. These results will be included in the final version.
>
> Please refer to the results in the anonymous link
> https://github.com/Anonymous3244/LSN_review/blob/main/Reviewer_KZfX.md
>
> >**Q2:**  There is no comparison against neural operator approaches and against state-of-the-art approaches from numerical analysis.
>
> **A2:**  Thanks. We conducted experiments comparing LSN with neural operator approaches (e.g., FNO) and traditional numerical methods (e.g., FDM). Results show that LSN outperform other methods. We will include these comparisons in the final version.
>
> Please refer to the results in the anonymous link
> https://github.com/Anonymous3244/LSN_review/blob/main/Reviewer_KZfX.md
>
>
>
> >**Q3:**  No mention of computation times is given. Is the new approach at least comparable in computation time to previous approaches?
>
> **A3:**  Thanks. We will add a discussion of computation times in the final version. Despite the additional constraints, LSN achieves test error of $5\times10^{−3}$ in 999.05 seconds for the Black-Scholes equation ($r=0.2$, $\sigma=0.2$), compared to 2600.82 seconds for PINN.
>
> >**Q4:** I get that this is not a paper going deeper into SDEs. However, one should at least mention that applying the Feynman-Kac formula (e.g., form (3) to (4)) loses information. The deterministic PDE in (4) is just an expectation of the SDE in (3).
>
> **A4:** We agree with the reviewer that the Feynman-Kac formula leads to a deterministic PDE that represents the expectation of the SDE, and thus some information is lost in this transformation. However, we respectfully note that our focus is on solving the resulting PDE, which provides valuable insights into the average behavior of the SDE.  We will clarify this in the final version.
>
> >**Q5:** The paper gives many citations (which is obviously a good idea), but is usually imprecise in its citation.
>
> **A5:**  Thanks and addressed.
>
> >**Q6:**  While you cite state-of-the-art methods, you do not describe differences. In particular, (Akhound-Sadegh et al., 2024) looks very similar to your approach. There, I would like to see the differences be clearly described.
>
> **A6:**   Thanks and addressed. Our LSN could potentially preserve a broader range of symmetries compared to LPS by leveraging conservation law systems. These systems can incorporate additional symmetries [1-5], including potential symmetries [1-2], nonlocal symmetries [3-4], and even novel local symmetries [5].  By leveraging the system of conservation laws, the symmetry analysis of the original equation can be extended, unveiling previously undiscovered structures and solutions. We will incorporate detailed explanations in the final version.
>
> ---
> >**Minor Comments:**
> Thanks and addressed.
>
>
> [1] Bluman G. Use and construction of potential symmetries[J]. Mathematical and computer modelling, 1993, 18(10): 1-14.
>
> [2] Pucci E, Saccomandi G. Potential symmetries and solutions by reduction of partial differential equations[J]. Journal of Physics A: Mathematical and General, 1993, 26(3): 681.
>
> [3] Bluman G W, Reid G J, Kumei S. New classes of symmetries for partial differential equations[J]. Journal of Mathematical Physics, 1988, 29(4): 806-811.
>
> [4] Olver P J. Applications of Lie groups to differential equations[M]. Springer Science & Business Media, 1993.
>
> [5] Edelstein R M, Govinder K S. Conservation laws for the Black–Scholes equation[J]. Nonlinear Analysis: Real World Applications, 2009, 10(6): 3372-3380.

---

> > ### Comment · Reviewer_KZfX · 2025-03-10
> > **Reply to Rebuttal**
> >
> > I thank the authors for the reply. My concerns are addressed and it seems to me the concerns raised by the other reviewers are addressed as well.
> >
> > It might sitll be helpful to add an experiment for a third PDE for the camera ready version.

---

> ### Author Response · Authors · 2025-03-14
> **Thanks! Further experiments.**
>
> Thanks very much for your kind support!
>
> Following your suggestion, we further conducted experiments for a third PDE (Maxwellian tails model: $u_{xt}+u_x+u^2=0$).  LSN consistently outperforms other methods in terms of test error. These results will be included in the final version.
>
> Please refer to the results in the anonymous link
> https://github.com/Anonymous3244/LSN_review/blob/main/Reviewer_KZfX.md

---

### Review · Reviewer_eWZt · 2025-02-25

**Summary Of Contributions:**

The paper proposes a new loss term, i.e. Lie symmetry risk, based on conservation laws derived from Lie point symmetries of PDEs for PINNs. The Lie symmetry risk is added to standard PINN losses to ensure the neural PDE solver learns solutions that respect the underlying symmetry of the PDE system. Compared to previous works incorporating Lie symmetry into PINN learning by optimizing the deviation from Lie's infinitesimal criterion (Akhound-Sadegh et al., 2024), the Lie symmetry risk used in this paper seems more indirect, since it requires the additional step of deriving the conservation laws from the Lie symmetries. The method is applied to the specific scenario of modeling the price of a call option in terms of the price of the underlying asset and time. Experimental evidence shows that the proposed loss term results in better prediction of PDE solutions evaluated on the test data points.

**Audience:**

Yes

**Claims And Evidence:**

Yes

**Requested Changes:**

* Add some intuitive explanation in the experiment section on why the proposed method works better than LPS (Akhound-Sadegh et al., 2024).
* In Figure 2, change the label on the Y-axis to "t".
* To demonstrate the overall effectiveness of all PINN-based methods, can you also include an error measure and an error plot for a dummy baseline, e.g. a second-order approximation of the option price using the Greeks?
* Add more details in Section 4.1 on how the conservation law corresponding to $G_2$ is derived, and on the reason why the specific one-parameter subgroup $G_2$ is chosen.
* Section 4.2 can be made more concise since it only involves the standard procedure of creating the empirical risk for optimization.
* Consider including more references in the Lie symmetries subsection. The authors may also find the following papers relevant as they all aim to incorporate Lie symmetries in learning for differential equation systems.
    * Brandstetter, J., Welling, M., and Worrall, D. E. Lie point symmetry data augmentation for neural pde solvers. In International Conference on Machine Learning, pp. 2241–2256. PMLR, 2022.
    * Otto, S. E., Zolman, N., Kutz, J. N., and Brunton, S. L. A unified framework to enforce, discover, and promote symmetry in machine learning, 2023.
    * Dalton, D., Husmeier, D., and Gao, H. Physics and lie symmetry informed gaussian processes. In Forty-first International Conference on Machine Learning, 2024.
    * Yang, J., Rao, W., Dehmamy, N., Walters, R., and Yu, R. Symmetry-informed governing equation discovery. In Advances in Neural Information Processing Systems (NeurIPS), 2024.

**Strengths And Weaknesses:**

Strength:
* The proposed solution for incorporating Lie symmetries of PDEs is simple and straightforward, though deriving the exact form of the Lie symmetry risk requires some prior knowledge.
* There are detailed descriptions of the experiment setup, and the results look promising for the specific case of modeling the price of a call option.

Weakness:
* The application of the proposed method is restricted to a single PDE system.
* Despite the extensive experiments, there is no intuitive explanation of why the proposed method could work better than LPS (Akhound-Sadegh et al., 2024), since both methods use loss terms based on Lie symmetries, only the exact forms of the loss terms differ.
* The derivation of the conservation law is not included in the main text. I think this is an important step of the method, and it would be better if some key points of the relevant discussion in Appendix A.3-5 could be moved to the main text.
* There are some missing references about exploiting Lie point symmetries for learning in differential equation systems.

---

> ### Author Response · Authors · 2025-03-10
>
> >**Q1:**   The application of the proposed method is restricted to a single PDE system.
>
> **A1:**  Thanks for the suggestion. We conducted further experiments to include the Vasicek model ($\frac{\partial u}{\partial t} +\alpha \frac{\partial u}{\partial x^2}+\lambda(\beta-x)\frac{\partial u}{\partial x} +\gamma x u=0$) and other KDV equations($u_t=u_{xxx}+uu_x$). LSN consistently outperforms others in terms of test error. These results will be included in the final version.
>
> Please refer to the results in the anonymous link
>  https://github.com/Anonymous3244/LSN_review/blob/main/Reviewer_eWZt.md
>
>
> >**Q2:** Despite the extensive experiments, there is no intuitive explanation of why the proposed method could work better than LPS (Akhound-Sadegh et al., 2024), since both methods use loss terms based on Lie symmetries, only the exact forms of the loss terms differ.
>
> **A2:**  Thanks. Our LSN could potentially preserve a broader range of symmetries compared to LPS by leveraging conservation law systems. These systems can incorporate additional symmetries [1-5], including potential symmetries [1-2], nonlocal symmetries [3-4], and even novel local symmetries [5].  By leveraging the system of conservation laws, the symmetry analysis of the original equation can be extended, unveiling previously undiscovered structures and solutions. We will incorporate detailed explanations in the final version.
>
> ---
>
> >**Q3:**  The derivation of the conservation law is not included in the main text. I think this is an important step of the method, and it would be better if some key points of the relevant discussion in Appendix A.3-5 could be moved to the main text.
>
> **A3:**  Thanks and addressed. We will move the conservation law derivation from Appendix A.3-5 to the main text.
>
>
>
> >**Q4:** There are some missing references about exploiting Lie point symmetries for learning in differential equation systems.
>
> **A4:**  Thanks. We will add Brandstetter et al. (2022) [6], Otto et al. (2023) [7], Dalton et al. (2024) [8], and Yang et al. (2024) [9].
>
>
> >**Q5:** Add some intuitive explanation in the experiment section on why the proposed method works better than LPS (Akhound-Sadegh et al., 2024).
>
> **A5:** Thanks. Please refer to A2.
>
> >**Q6:** In Figure 2, change the label on the Y-axis to "t".
>
> **A6:**  Thanks and addressed.
>
> >**Q7:**   To demonstrate the overall effectiveness of all PINN-based methods, can you also include an error measure and an error plot for a dummy baseline, e.g., a second-order approximation of the option price using the Greeks?
>
> **A7:**  Thanks and addressed. We plotted error curves with respect to data volume. The slope of the curves reflect data utilization efficiency: steeper slopes indicate higher efficiency of data utilization and lower sample complexity. Compared to PINN, LSN does not increase sampling complexity. While both PINN and LSN are less sensitive to data volume than finite differences, LSN achieves lower overall error.
>
> Please refer to the error curves in the anonymous link
> https://github.com/Anonymous3244/LSN_review/blob/main/Reviewer_eWZt.md.
>
> >**Q8:.**   Add more details in Section 4.1 on how the conservation law corresponding to G2 is derived, and on the reason why the specific one-parameter subgroup G2 is chosen
>
> **A8:**  Thanks. We will include the derivation of the conservation law corresponding to G2 in Section 4.1. We chose G2 because it is simple, allowing us to clearly demonstrate the core ideas of our method. We have also included additional experiments for Vasicek equation in the Appendix A.3 using different operators to validate the method’s effectiveness.
>
> Please refer to the results in the anonymous link
>  https://github.com/Anonymous3244/LSN_review/blob/main/Reviewer_eWZt.md
>
>
> >**Q9:**  Section 4.2 can be made more concise since it only involves the standard procedure of creating the empirical risk for optimization.
>
> **A9:**  Thanks and addressed.
>
> >**Q10:**  Consider including more references in the Lie symmetries subsection.
>
> **A10:**  Thanks. Please refer to A4.

---

> ### Author Response · Authors · 2025-03-10
>
> [1] Bluman G. Use and construction of potential symmetries[J]. Mathematical and computer modelling, 1993, 18(10): 1-14.
>
> [2] Pucci E, Saccomandi G. Potential symmetries and solutions by reduction of partial differential equations[J]. Journal of Physics A: Mathematical and General, 1993, 26(3): 681.
>
> [3] Bluman G W, Reid G J, Kumei S. New classes of symmetries for partial differential equations[J]. Journal of Mathematical Physics, 1988, 29(4): 806-811.
>
> [4] Olver P J. Applications of Lie groups to differential equations[M]. Springer Science & Business Media, 1993.
>
> [5] Edelstein R M, Govinder K S. Conservation laws for the Black–Scholes equation[J]. Nonlinear Analysis: Real World Applications, 2009, 10(6): 3372-3380.
>
> [6] Brandstetter, J., Welling, M., and Worrall, D. E. Lie point symmetry data augmentation for neural pde solvers. In International Conference on Machine Learning, pp. 2241–2256. PMLR, 2022.
>
> [7] Otto, S. E., Zolman, N., Kutz, J. N., and Brunton, S. L. A unified framework to enforce, discover, and promote symmetry in machine learning, 2023.
>
> [8] Dalton, D., Husmeier, D., and Gao, H. Physics and lie symmetry informed gaussian processes. In Forty-first International Conference on Machine Learning, 2024.
>
> [9] Yang, J., Rao, W., Dehmamy, N., Walters, R., and Yu, R. Symmetry-informed governing equation discovery. In Advances in Neural Information Processing Systems (NeurIPS), 2024.

---

> > ### Author Response · Authors · 2025-04-09
> > **Looking forward to your reply!**
> >
> > Thanks again for your time and efforts in reviewing our paper. We sincerely hope all your concerns have been cleared. We are happy to answer any further question!

---

> > > ### Comment · Reviewer_eWZt · 2025-04-09
> > >
> > > Thank you for your reply. My concern has been addressed and I don’t have further questions.

---

> > > > ### Author Response · Authors · 2025-04-09
> > > > **Thanks!**
> > > >
> > > > Thank you very much for your kind support!

---

### Review · Reviewer_QiWW · 2025-02-25

**Summary Of Contributions:**

This paper proposes the Lie Symmetry Network (LSN), a Physics-Informed Neural Network (PINN) designed to characterize and preserve the Lie symmetries of differential equations. LSN enhances the standard PINN loss by incorporating an L2 error term derived from Lie-symmetry-based conservation laws (or Lie symmetry risks), ensuring that learned solutions adhere to fundamental invariance properties. Empirical results demonstrate that LSN achieves state-of-the-art performance on the Black-Scholes equation and consistently outperforms Lie Point Symmetry (LPS) methods across various settings.

**Audience:**

Yes

**Broader Impact Concerns:**

.

**Claims And Evidence:**

Yes

**Requested Changes:**

Summarization of the requests described in Weakness section:

- PINN, LPS, LSN experiments on $\sigma=0.05, 0.2, 0.5, 0.8$, $r=0.02, 0.05, 0.1, 0.2$ with $\lambda_4 = 1e-3, 1e-4, 1e-5$, Total iterations of 200K.

- Could authors provide the reason why LSN performs better than LPS?

- Could authors include experiments on European and American option pricing?

**Strengths And Weaknesses:**

**Strength**
- The paper is well-organized and well-written, making it easy to follow.
- LSN demonstrates a significant performance improvement over existing methods.

**Weakness**

Many prior works have incorporated additional residual objectives into PINN to enforce specific constraints. These works significantly improves the performance for the task in the specific domain. I was convinced that LSN effectively enhances PINN for the Black-Scholes (BS) equation and achieves superior performance. However, I believe some aspects require further verification:

- **Comparative Analysis for LPS** The comparison with LPS is currently limited to settings where $\sigma\approx 0.5$ and $r \approx 0.1$. A more comprehensive evaluation across a wider range of parameter values would strengthen the analysis. Specifically, could authors provide results for PINN variants, LPS, and LSN across $\sigma=0.05, 0.1, 0.2, 0.5$, $r=0.01, 0.02, 0.05, 0.1, 0.2$ (total 20 experiments for each methods)?  These values are particularly relevant given that, in real-world scenarios ($r\approx 0.2$. $\sigma=0.2$).

- Moreover, I believe LPS should be trained with various $\lambda_4$ for the fair comparison. I suggest using $\lambda_4 = 1e-3, 1e-4, 1e-5$.

- The number of training iterations varies across experiments in the BS setting. Would it be possible to standardize this to 200K iterations for all methods to ensure comparability?

- **Interpretability of LSN's advantage**: Both LPS and LSN incorporate residual loss terms related to Lie symmetry. Could authors provide an intuitive or practical explanation, possibly supported by toy experiments, to illustrate why LSN performs better?

- **Additional Experiments on Different Dataset**: Could authors include experiments on European and American option pricing following [1]? This would further demonstrate the applicability of LSN beyond the Black-Scholes equation.

[1] Ashish Dhiman, Yibei Hu, Physics Informed Neural Network for Option Pricing, 2023.

---

> ### Author Response · Authors · 2025-03-10
>
> >**Q1:**  Comparative Analysis for LPS
> The comparison with LPS is currently limited to settings where $\sigma=0.5$ and$r=0.1$. A more comprehensive evaluation across a wider range of parameter values would strengthen the analysis. Specifically, could authors provide results for PINN variants, LPS, and LSN across $\sigma=0.05, 0.1, 0.2, 0.5$, $r=0.01, 0.02, 0.05, 0.1, 0.2$ (total 20 experiments for each method)? These values are particularly relevant given that, in real-world scenarios ( $\sigma=0.2$, $r=0.2$).
>
> **A1:**  Thanks. We agree, but please note – each experiment needs to run for 200,000 steps, thus conducting 20 experiments for each method is time-prohibitive within the two-week timeframe. Instead, we have conducted experiments with typical parameters covering both small and large differences in $\sigma$ and $r$ to ensure a representative evaluation -$(r, \sigma)=\{(0.05, 0.2), (0.2, 0.2), (0.2, 0.05)\}$. The results show consistent performance of LSN across these parameters, further validating its robustness.
>
> Please refer to the results in the anonymous link
> https://github.com/Anonymous3244/LSN_review/blob/main/Reviewer_QiWW.md
>
>
> >**Q2:** Moreover, I believe LPS should be trained with various $\lambda_4$ for fair comparison. I suggest using $\lambda_4=1e−3, 1e−4, 1e−5$.
>
> **A2:**  Thanks. To ensure a fair comparison, we have meticulously fine-tuned LPS in Figure 7 of the manuscript. Specifically, we conducted experiments over a defined range of values for $\lambda_4$, including 10, 1, 1e-1, 1e-2, 1e−3, and 1e−4.
>
> Please refer to the results in the anonymous link
> https://github.com/Anonymous3244/LSN_review/blob/main/Reviewer_QiWW.md
>
> >**Q3:**  The number of training iterations varies across experiments in the BS setting. Would it be possible to standardize this to 200K iterations for all methods to ensure comparability?
>
> **A3:**  Thank you for your observation. Regarding the experimental setup, we would like to clarify that while some experiments, such as those in Figure 5, were conducted with 80,000 training steps where most methods had already converged.
>
> >**Q4:**  Interpretability of LSN's advantage: Both LPS and LSN incorporate residual loss terms related to Lie symmetry. Could authors provide an intuitive or practical explanation, possibly supported by toy experiments, to illustrate why LSN performs better?
>
> **A4:**  Thanks. Our LSN could potentially preserve a broader range of symmetries compared to LPS by leveraging conservation law systems. These systems can incorporate additional symmetries [1-5], including potential symmetries [1-2], nonlocal symmetries [3-4], and even novel local symmetries [5].  By leveraging the system of conservation laws, the symmetry analysis of the original equation can be extended, unveiling previously undiscovered structures and solutions. We will incorporate detailed explanations in the final version.
>
> >**Q5:** Additional Experiments on Different Dataset: Could authors include experiments on European and American option pricing following [1]? This would further demonstrate the applicability of LSN beyond the Black-Scholes equation.
>
> **A5:**  Thanks and addressed. We have added experiments on European option pricing using real-world financial data. The results demonstrate that LSN performs well in this setting, further validating its applicability beyond the Black-Scholes equation. These results will be included in the final version.
>
> Please refer to the results in the anonymous link
> https://github.com/Anonymous3244/LSN_review/blob/main/Reviewer_QiWW.md
>
>
> [1] Bluman G. Use and construction of potential symmetries[J]. Mathematical and computer modelling, 1993, 18(10): 1-14.
>
> [2] Pucci E, Saccomandi G. Potential symmetries and solutions by reduction of partial differential equations[J]. Journal of Physics A: Mathematical and General, 1993, 26(3): 681.
>
> [3] Bluman G W, Reid G J, Kumei S. New classes of symmetries for partial differential equations[J]. Journal of Mathematical Physics, 1988, 29(4): 806-811.
>
> [4] Olver P J. Applications of Lie groups to differential equations[M]. Springer Science & Business Media, 1993.
>
> [5] Edelstein R M, Govinder K S. Conservation laws for the Black–Scholes equation[J]. Nonlinear Analysis: Real World Applications, 2009, 10(6): 3372-3380.

---

> > ### Comment · Reviewer_QiWW · 2025-03-28
> >
> > I thank the authors for the reply and for the additional experiments. My concerns are adequately addressed.

---

> ### Author Response · Authors · 2025-03-29
> **Thanks!**
>
> Thank you very much for your kind support!

---

### Decision · Action_Editor_YgNm · 2025-05-01

**Recommendation:** Accept with minor revision

**Comment:**

Overall, the reviewers have appreciated the work. Their concerns were mostly resolved during the feedback phase, and all of them have acknowledged the same. However, a few minor concerns still remain, which should be addressed in the paper.

1. It might still be helpful to add an experiment for a third PDE for the camera-ready version. A reviewer has suggested this.

2. A reviewer raises critical concerns about the motivation behind using the conservation loss instead of the established Lie symmetry loss term. It might be a good idea to explain this better in the paper. Some additional experiments in this regard would be useful. I am not sure how feasible this might be, but do take a stab at them.

Based on the reviews and discussions, I believe the paper is suitable for publication in TMLR with minor revisions.

**Audience:**

Yes.

**Claims And Evidence:**

Yes.

---

> ### Author Response · Authors · 2025-05-09
> **Thanks; all concerns will be duly addressed**
>
> We sincerely appreciate this decision and the efforts of the action editor and reviewers for reviewing our paper. All your remained concerns will be duly addressed.
>
> Cheers,
> the authors.

---

> ### Author Response · Authors · 2025-05-29
> **Minor revision completed - additional experiments and explainations**
>
> Thanks again for the time managing the review of this paper!
>
> Your concerns have been duly addressed in the final version uploaded.
>
> To your concerns:
>
> 1. We added experiments for Maxwellian Tails model, in addition to the presented Black-Scholes equation, Vašiček equation, and KdV equation; please kindly referred to Section 5.2. The results suggest that our approach significantly outperforms all existing PINN methods.
>
> 2. We added an explanation below on the motivation of using the conservation loss instead of the established Lie symmetry loss term, in page 13, and as follows,
>
> “Our conservation law-based approach can capture a broader range of symmetries, compared with using a pre-fixed symmetry operator. More specifically, our approach preserves Lie symmetry operators G₂, l(t), and g(t) (see Equation (7)), while exsiting method can only cover G₂. We experimentally validated the advances in Figure 9.”
>
> We also conducted additional experiments on real financial data, the OptionMetrics dataset based on the Nasdaq 100 index, to verify our approach, which suggests that our method outperformes existing methods.